# LAMBDANETWORKS: MODELING LONG-RANGE INTERACTIONS WITHOUT ATTENTION

**Irwan Bello**
Google Research, Brain team
ibello@google.com

## ABSTRACT

We present lambda layers – an alternative framework to self-attention – for capturing long-range interactions between an input and structured contextual information (e.g. a pixel surrounded by other pixels). Lambda layers capture such interactions by transforming available contexts into linear functions, termed lambdas, and applying these linear functions to each input separately. Similar to linear attention, lambda layers bypass expensive attention maps, but in contrast, they model both content and *position-based* interactions which enables their application to large structured inputs such as images. The resulting neural network architectures, *LambdaNetworks*, significantly outperform their convolutional and attentional counterparts on ImageNet classification, COCO object detection and instance segmentation, while being more computationally efficient. Additionally, we design LambdaResNets, a family of hybrid architectures across different scales, that considerably improves the speed-accuracy tradeoff of image classification models. LambdaResNets reach excellent accuracies on ImageNet while being 3.2 - 4.4x faster than the popular EfficientNets on modern machine learning accelerators. In large-scale semi-supervised training with an additional 130M pseudo-labeled images, LambdaResNets achieve up to 86.7% ImageNet accuracy while being 9.5x faster than EfficientNet NoisyStudent and 9x faster than a Vision Transformer with comparable accuracies[1].

## 1 INTRODUCTION

Modeling long-range dependencies in data is a central problem in machine learning. Self-attention (Bahdanau et al., 2015; Vaswani et al., 2017) has emerged as a popular approach to do so, but the costly memory requirement of self-attention hinders its application to long sequences and multidimensional data such as images[2]. *Linear (or efficient)* attention mechanisms (Katharopoulos et al., 2020; Choromanski et al., 2020) offer a scalable remedy for high memory usage but fail to model internal data structure, such as *relative* distances between pixels or edge relations between nodes in a graph.

This work addresses both issues. We propose *lambda layers* which model long-range interactions between a query and a *structured* set of context elements at a reduced memory cost. Lambda layers transform each available context into a linear function, termed a *lambda*, which is then directly applied to the corresponding query. Whereas self-attention defines a similarity kernel between the query and the context elements, a lambda layer instead summarizes contextual information into a fixed-size linear function (i.e. a matrix), thus bypassing the need for memory-intensive attention maps. This difference is illustrated in Figure 1.

Lambda layers are versatile and can be implemented to model both content-based and *position-based* interactions in global, local or masked contexts. The resulting neural networks, *LambdaNetworks*, are computationally efficient, model long-range dependencies at a small memory cost and can therefore be applied to large structured inputs such as high resolution images.

---

[1] An updated version of this paper can be found on arXiv.

[2] For example, applying a single multi-head attention layer to a batch of 128 64x64 input images with 8 heads requires 64GB of memory, which is prohibitive in practice.

GLOBAL CONTEXT

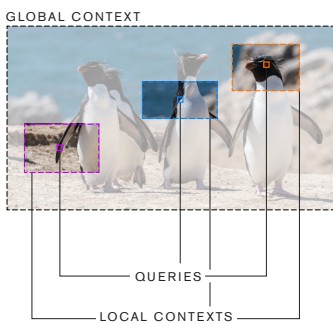
QUERIES
LOCAL CONTEXTS

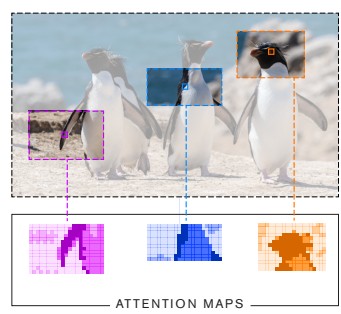
ATTENTION MAPS

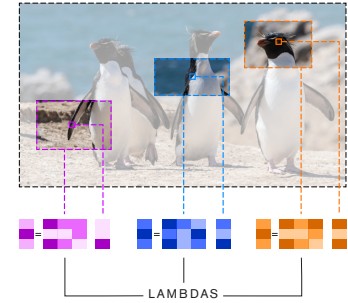
LAMBDAS

Figure 1: **Comparison between self-attention and lambda layers**. (**Left**) An example of 3 queries and their local contexts within a global context. (**Middle**) Self-attention associates each query with an attention distribution over its context. (**Right**) The lambda layer transforms each context into a linear function lambda that is applied to the corresponding query.

We evaluate LambdaNetworks on computer vision tasks where works using self-attention are hindered by large memory costs (Wang et al., 2018; Bello et al., 2019), suffer impractical implementations (Ramachandran et al., 2019), or require vast amounts of data (Dosovitskiy et al., 2020). In our experiments spanning ImageNet classification, COCO object detection and instance segmentation, LambdaNetworks significantly outperform their convolutional and attentional counterparts, while being more computationally efficient and faster than the latter. We summarize our contributions:

- **Lambda layers**: a class of layers, that model content-based and position-based interactions without materializing attention maps. Lambda layers offer a unifying view of channel, spatial and linear attention (Appendix D.4). Some of our observations, such as the computational benefits of a multi-query formulation, extend to linear attention. Lambda layers are easily implemented with einsum operations and convolution kernels, operations with efficient implementations on modern machine learning accelerators.

- Lambda layers significantly outperform their convolution and attention counterparts on the ImageNet classification task while being more computationally efficient. For example, simply replacing the 3x3 convolutions in the bottleneck blocks of the ResNet-50 architecture (He et al., 2016) with lambda layers yields a +1.5% top-1 ImageNet accuracy improvement while reducing parameters by 40% (Section 5.1).

- Lambda layers achieve considerable computational benefits, both in latency and memory requirements, over multiple self-attention alternatives, including local and axial attention (Ramachandran et al., 2019; Wang et al., 2020a). When used in a ResNet-50 architecture at image resolution 224, lambda layers reduce memory consumption by ∼200x compared to *global* attention (∼7x compared to *axial* attention) while being ∼3.7x faster than *local* attention (Section 5.2).

- A study of *hybrid convolution-lambda models* as a means to maximize the speed-accuracy tradeoff (Section 5.3). Hybrid designs that first employ convolutions at the highest resolution and lambda layers in intermediate to low resolutions achieve the best speed-accuracy tradeoff.

- **LambdaResNets**: a family of hybrids based on the training and scaling strategies recommended in Bello et al. (2021). LambdaResNets achieve up to a **4.4x** speed-up over EfficientNets on ImageNet, while being more memory-efficient. LambdaResNets can also be designed for parameter or flops efficiency. For example, a LambdaResNet with 42M parameters achieves 84.3% top-1 ImageNet accuracy at image resolution 320 (Section E.4).

- In large-scale semi-supervised training with an additional 130M pseudo-labeled images, LambdaResNets achieve up to **86.7%** top-1 ImageNet accuracy while being **9.5x** faster than EfficientNet *NoisyStudent* (Xie et al., 2020) and **9x** faster than a *Vision Transformer* (Dosovitskiy et al., 2020) with comparable accuracies (Section 5.3).

- An evaluation of LambdaResNets on COCO object detection and instance segmentation using Mask-RCNN (He et al., 2017). LambdaResNet backbones yield consistent gains across all metrics on both tasks (e.g. +1.8% mAP improvement for detecting small objects).

| |
|---|
| A **content-based** interaction considers the content of the context but ignores the relation between the query position and the context (e.g. relative distance between two pixels). A **position-based** interaction considers the relation between the query position and the context position. |

Table 1: **Definition of content-based vs position-based interactions.**

## 2  MODELING LONG-RANGE INTERACTIONS

In this section, we formally define queries, contexts and interactions. Starting from first principles, we motivate keys and relative position embeddings as a requirement for capturing structured interactions between queries and their contexts. We then show that lambda layers arise as an alternative to attention mechanisms for capturing long-range interactions.

**Notation.**   We denote scalars, vectors and tensors using lower-case, bold lower-case and bold upper-case letters, *e.g.*, $n$, $\boldsymbol{x}$ and $\boldsymbol{X}$. We denote $|n|$ the cardinality of a set whose elements are indexed by $n$. We denote $\boldsymbol{x}_n$ the $n$-th row of $\boldsymbol{X}$. We denote $x_{ij}$ the $|ij|$ elements of $\boldsymbol{X}$. When possible, we adopt the terminology of self-attention to ease readability and highlight differences.

### 2.1  MOTIVATING QUERIES, KEYS, POSITION EMBEDDINGS AND VALUES

**Defining queries and contexts.**   Let $\mathcal{Q} = \{(\boldsymbol{q}_n, n)\}$ and $\mathcal{C} = \{(\boldsymbol{c}_m, m)\}$ denote structured collections of vectors, respectively referred to as the *queries* and the *context*. Each query $(\boldsymbol{q}_n, n)$ is characterized by its content $\boldsymbol{q}_n \in \mathbb{R}^{|k|}$ and *position* $n$. Similarly, each context element $(\boldsymbol{c}_m, m)$ is characterized by its *content* $\boldsymbol{c}_m$ and its position $m$ in the context. The $(n, m)$ pair may refer to any pairwise relation between structured elements, e.g. relative distances between pixels or edges between nodes in a graph.

**Defining interactions.**   We consider the general problem of mapping a query $(\boldsymbol{q}_n, n)$ to an output vector $\boldsymbol{y}_n \in \mathbb{R}^{|v|}$ given the context $\mathcal{C}$ with a function $\boldsymbol{F} : ((\boldsymbol{q}_n, n), \mathcal{C}) \mapsto \boldsymbol{y}_n$. Such a function may act as a layer in a neural network when processing structured inputs.

We refer to $(\boldsymbol{q}_n, \boldsymbol{c}_m)$ interactions as *content-based* and $(\boldsymbol{q}_n, (n, m))$ interactions as *position-based*. We note that while *absolute* positional information is sometimes directly added to the query (or context element) content[3], we consider this type of interaction to be content-based as it ignores the *relation* $(n, m)$ between the query and context element positions.

**Introducing keys and relative position embeddings to capture long-range interactions.**   In the context of deep learning, we prioritize fast batched linear operations and use dot-product operations as our interactions. This motivates introducing vectors that can interact with the queries via a dot-product operation and therefore have the same dimension as the queries. In particular, content-based interactions $(\boldsymbol{q}_n, \boldsymbol{c}_m)$ require a $|k|$-dimensional vector that depends on $\boldsymbol{c}_m$, commonly referred to as the key $\boldsymbol{k}_m$. Conversely, position-based interactions $(\boldsymbol{q}_n, (n, m))$ require a relative position embedding $\boldsymbol{e}_{nm} \in \mathbb{R}^{|k|}$ (Shaw et al., 2018). As the query/key depth $|k|$ and context spatial dimension $|m|$ are not in the output $\boldsymbol{y}_n \in \mathbb{R}^{|v|}$, these dimensions need to be contracted as part of the layer computations. Therefore

> *Every layer capturing long-range interactions can be characterized based on whether it contracts (1) the query depth or (2) the context positions first.*

### 2.2  ATTENTION VS LAMBDA LAYERS.

**(1) Attention layers.**   Contracting the query depth first creates a similarity kernel (the attention map) between the query and context elements and is known as the attention operation. As the number of context positions $|m|$ grows larger and the input and output dimensions $|k|$ and $|v|$ remain fixed, one may hypothesize that computing attention maps become wasteful, given that the layer output is a vector of comparatively small dimension $|v| \ll |m|$.

**(2) Lambda layers.**   Instead, it may be more efficient to simply map each query to its output as $\boldsymbol{y}_n = F((\boldsymbol{q}_n, n), \mathcal{C}) = \boldsymbol{\lambda}(\mathcal{C}, n)(\boldsymbol{q}_n)$ for some *linear* function $\boldsymbol{\lambda}(\mathcal{C}, n) : \mathbb{R}^{|k|} \to \mathbb{R}^{|v|}$. In this

---

[3]This approach is often used in natural language processing tasks (Vaswani et al., 2017) but has had limited success in the visual domain where relative position information between pixels is crucial (Bello et al., 2019).

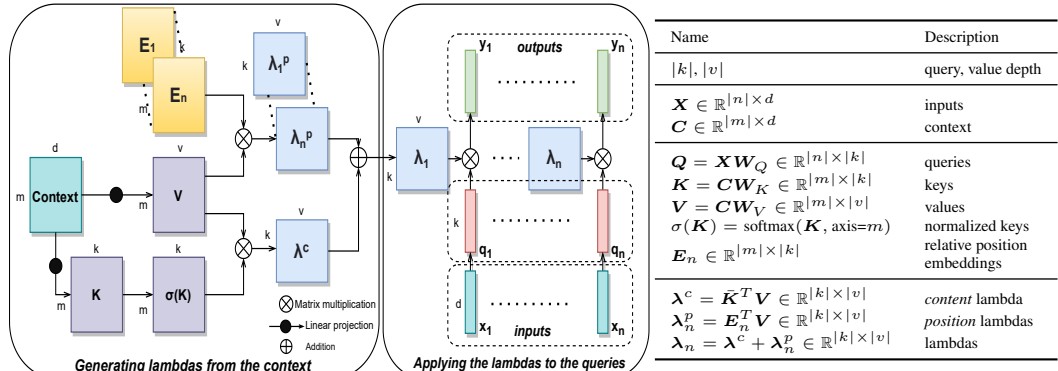

Figure 2: **Computational graph of the lambda layer**. Contextual information for query position $n$ is summarized into a lambda $\boldsymbol{\lambda}_n \in \mathbb{R}^{|k| \times |v|}$. Applying the lambda dynamically distributes contextual features to produce the output as $\boldsymbol{y}_n = \boldsymbol{\lambda}_n^T \boldsymbol{q}_n$. This process captures content-based and position-based interactions without producing attention maps.

scenario, the context is aggregated into a fixed-size linear function $\boldsymbol{\lambda}_n = \boldsymbol{\lambda}(\mathcal{C}, n)$. Each $\boldsymbol{\lambda}_n$ acts as a small linear function[4] that exists independently of the context (once computed) and is discarded after being applied to its associated query $\boldsymbol{q}_n$.

## 3 LAMBDA LAYERS

### 3.1 LAMBDA LAYER: TRANSFORMING CONTEXTS INTO LINEAR FUNCTIONS.

A *lambda layer* takes the inputs $\boldsymbol{X} \in \mathbb{R}^{|n| \times d_{in}}$ and the context $\boldsymbol{C} \in \mathbb{R}^{|m| \times d_c}$ as input and generates linear function lambdas that are then applied to the queries, yielding outputs $\boldsymbol{Y} \in \mathbb{R}^{|n| \times d_{out}}$. Without loss of generality, we assume $d_{in} = d_c = d_{out} = d$. As is the case with *self*-attention, we we may have $\boldsymbol{C} = \boldsymbol{X}$. In the rest of this paper, we focus on a *specific instance of a lambda layer* and show that it captures long-range content and position-based interactions without materializing attention maps. Figure 2 presents the computational graph of the lambda layer.

We first describe the lambda layer when applied to a *single query* $(\boldsymbol{q}_n, n)$.

**Generating the contextual lambda function.** We wish to generate a linear function $\mathbb{R}^{|k|} \to \mathbb{R}^{|v|}$, i.e. a matrix $\boldsymbol{\lambda_n} \in \mathbb{R}^{|k| \times |v|}$. The lambda layer first computes *keys* $\boldsymbol{K}$ and *values* $\boldsymbol{V}$ by linearly projecting the context, and keys are normalized across context positions via a softmax operation yielding normalized keys $\bar{\boldsymbol{K}}$. The $\boldsymbol{\lambda}_n$ matrix is obtained by using the normalized keys $\bar{\boldsymbol{K}}$ and position embeddings $\boldsymbol{E}_n$ to aggregate the values $\boldsymbol{V}$ as

$$\boldsymbol{\lambda}_n = \sum_m (\bar{\boldsymbol{k}}_m + \boldsymbol{e}_{nm}) \boldsymbol{v}_m^T = \underbrace{\bar{\boldsymbol{K}}^T \boldsymbol{V}}_{\text{content lambda}} + \underbrace{\boldsymbol{E}_n^T \boldsymbol{V}}_{\text{position lambda}} \in \mathbb{R}^{|k| \times |v|} \tag{1}$$

where we also define the *content lambda* $\boldsymbol{\lambda}^c$ and *position lambda* $\boldsymbol{\lambda}_n^p$.

- The *content lambda* $\boldsymbol{\lambda}^c$ is shared across all query positions $n$ and is invariant to permutation of the context elements. It encodes how to transform the query $\boldsymbol{q}_n$ solely based on the context content.

- The *position lambda* $\boldsymbol{\lambda}_n^p$ depends on the query position $n$ via the position embedding $\boldsymbol{E}_n$. It encodes how to transform the query $\boldsymbol{q}_n$ based on the context elements $\boldsymbol{c}_m$ and their *relative positions* to the query $(n, m)$.

**Applying lambda to its query.** The query $\boldsymbol{q}_n \in \mathbb{R}^{|k|}$ is obtained from the input $\boldsymbol{x}_n$ via a learned linear projection and the output of the lambda layer is obtained as

$$\boldsymbol{y}_n = \boldsymbol{\lambda}_n^T \boldsymbol{q}_n = (\boldsymbol{\lambda}^c + \boldsymbol{\lambda}_n^p)^T \boldsymbol{q}_n \in \mathbb{R}^{|v|}. \tag{2}$$

---

[4]This mechanism is reminiscent of functional programming and $\lambda$-calculus which motivates the lambda terminology.

**Interpretation of lambda layers.** The columns of the $\boldsymbol{\lambda}_n \in \mathbb{R}^{|k| \times |v|}$ matrix can be viewed as a fixed-size set of $|k|$ contextual features. These contextual features are aggregated based on the context's content (content-based interactions) and structure (position-based interactions). Applying the lambda then dynamically distributes these contextual features based on the query to produce the output as $\boldsymbol{y}_n = \sum_k q_{nk} \boldsymbol{\lambda}_{nk}$. This process captures content and position-based interactions without producing attention maps and can be viewed as an *efficient relative attention* mechanism.

**Normalization.** One may modify Equations 1 and 2 to include non-linearities or normalization operations. Our experiments indicate that applying batch normalization (Ioffe & Szegedy, 2015) after computing the queries and the values is helpful.

## 3.2 A MULTI-QUERY FORMULATION TO REDUCE COMPLEXITY.

**Complexity analysis.** For a batch of $|b|$ examples, each containing $|n|$ inputs, the number of arithmetic operations and memory footprint required to apply our lambda layer are respectively $\Theta(bnmkv)$ and $\Theta(knm + bnkv)$. We still have a quadratic memory footprint with respect to the input length due to the $\boldsymbol{e}_{nm}$ relative position embeddings. However this quadratic term does not scale with the batch size as is the case with the attention operation which produces *per-example* attention maps. In practice, the hyperparameter $|k|$ is set to a small value (such as $|k|$=16) and we can process large batches of large inputs in cases where attention cannot (see Table 4). Additionally, position embeddings can be shared across lambda layers to keep their $\Theta(knm)$ memory footprint constant - whereas the memory footprint of attention maps scales with the number of layers[5].

```
def lambda_layer(queries, keys, embeddings, values):
    """Multi-query lambda layer."""
    # b: batch, n: input length, m: context length,
    # k: query/key depth, v: value depth,
    # h: number of heads, d: output dimension.
    content_lambda = einsum(softmax(keys), values, 'bmk,bmv->bkv')
    position_lambdas = einsum(embeddings, values, 'nmk,bmv->bnkv')
    content_output = einsum(queries, content_lambda, 'bhnk,bkv->bnhv')
    position_output = einsum(queries, position_lambdas, 'bhnk,bnkv->bnhv')
    output = reshape(content_output + position_output, [b, n, d])
    return output
```

Figure 3: **Pseudo-code for the multi-query lambda layer**. The position embeddings can be made to satisfy various conditions, such as translation equivariance, when computing positional lambdas (not shown). The lambda layer can be adapted to other tasks/modalities by adjusting the choice of embeddings (Section A.2).

**Multi-query lambda layers reduce time and space complexities.** Recall that the lambda layer maps inputs $\boldsymbol{x}_n \in \mathbb{R}^d$ to outputs $\boldsymbol{y}_n \in \mathbb{R}^d$. As presented in Equation 2, this implies that $|v|$=d. Small values of $|v|$ may therefore act as a bottleneck on the feature vector $\boldsymbol{y}_n$ but larger output dimensions $|v|$ can incur an excessively large computational cost given our $\Theta(bnmkv)$ and $\Theta(knm + bnkv)$ time and space complexities.

We propose to decouple the time and space complexities of our lambda layer from the output dimension $d$. Rather than imposing $|v|$=d, we create $|h|$ queries $\{\boldsymbol{q}_n^h\}$, apply the same lambda $\boldsymbol{\lambda}_n$ to each query $\boldsymbol{q}_n^h$, and concatenate the outputs as $\boldsymbol{y}_n = \text{concat}(\boldsymbol{\lambda}_n \boldsymbol{q}_n^1, \cdots, \boldsymbol{\lambda}_n \boldsymbol{q}_n^{|h|})$. We now have $|v|$=d/$|h|$, which reduces complexity by a factor of $|h|$. The number of *heads* $|h|$ controls the size of the lambdas $\boldsymbol{\lambda}_n \in \mathbb{R}^{|k| \times |d|/|h|}$ relative to the total size of the queries $\boldsymbol{q}_n \in \mathbb{R}^{|hk|}$.

We refer to this operation as a *multi-query* lambda layer and present an implementation using einsum[6] in Figure 3. The lambda layer is robust to $|k|$ and $|h|$ hyperparameter choices (see Appendix E.1), which enables flexibility in controlling its complexity. We use $|h|$=4 in most experi-

---

[5]Attention maps typically need to be stored for back-propagation (Kitaev et al., 2020).

[6]The einsum operation denotes general contractions between tensors of arbitrary dimensions. It is numerically equivalent to broadcasting its inputs to share the union of their dimensions, multiplying element-wise and summing across all dimensions not specified in the output.

| Operation | Head configuration | Interactions | Time complexity | Space complexity |
|---|---|---|---|---|
| Attention | multi-head | content-only | $\Theta(bnm(hk + d))$ | $\Theta(bhnm)$ |
| Relative attention | multi-head | content & position | $\Theta(bnm(hk + d))$ | $\Theta(bhnm)$ |
| Linear attention | multi-head | content-only | $\Theta(bnkd)$ | $\Theta(bkd)$ |
| Lambda layer | multi-query | content & position | $\Theta(bnmkd/h)$ | $\Theta(knm + bnkd/h)$ |
| Lambda convolution | multi-query | content & position | $\Theta(bnrkd/h)$ | $\Theta(kr + bnkd/h)$ |

Table 2: **Alternatives for capturing long-range interactions.** The lambda layer captures content and *position-based* interactions at a reduced memory cost compared to relative attention (Shaw et al., 2018; Bello et al., 2019). Using a multi-query lambda layer reduces complexities by a factor of $|h|$. Additionally, position-based interactions can be restricted to a local scope by using the lambda convolution which has linear complexity. $b$: batch size, $h$: number of heads/queries, $n$: input length, $m$: context length, $r$: local scope size, $k$: query/key depth, $d$: dimension output.

ments. We note that while this resembles the multi-head or multi-query (Shazeer, 2019)[7] attention formulation, the motivation is different. Using multiple queries in the attention operation increases representational power and complexity. In contrast, using multiple queries in the lambda layer *decreases* complexity and representational power (ignoring the additional queries).

**Extending the multi-query formulation to linear attention.** Finally, we point that our analysis extends to linear attention which can be viewed as a *content-only* lambda layer (see Appendix D.3 for a detailed discussion). We anticipate that the multi-query formulation can also bring computational benefits to linear attention mechanisms.

### 3.3 MAKING LAMBDA LAYERS TRANSLATION EQUIVARIANT.

Using relative position embeddings $\boldsymbol{e}_{nm}$ enables making explicit assumptions about the structure of the context. In particular, translation equivariance (i.e. the property that shifting the inputs results in an equivalent shift of the outputs) is a strong inductive bias in many learning scenarios. We obtain translation equivariance in position interactions by ensuring that the position embeddings satisfy $\boldsymbol{e}_{nm} = \boldsymbol{e}_{t(n)t(m)}$ for any translation $t$. In practice, we define a tensor of *relative* position embeddings $\boldsymbol{R} \in \mathbb{R}^{|r| \times |k|}$, where $r$ indexes the possible relative positions for all $(n, m)$ pairs, and reindex[8] it into $\boldsymbol{E} \in \mathbb{R}^{|n| \times |m| \times |k|}$ such that $\boldsymbol{e}_{nm} = \boldsymbol{r}_{r(n,m)}$.

### 3.4 LAMBDA CONVOLUTION: LOCAL CONTEXTS ON THE GRID.

Despite the benefits of long-range interactions, locality remains a strong inductive bias in many tasks. Using global contexts may prove noisy or computationally excessive. It may therefore be useful to restrict the scope of position interactions to a *local* neighborhood around the query position $n$ as is the case for local self-attention and convolutions. This can be done by zeroing out the relative embeddings for context positions $m$ outside of the desired scope. However, this strategy remains costly for large values of $|m|$ since the computations still occur - they are only being zeroed out.

**Lambda convolution** In the case where the context is arranged in a multidimensional grid, we can equivalently compute *positional lambdas* from local contexts by using a regular convolution. We term this operation the *lambda convolution*. A n-dimensional lambda convolution can be implemented using an n-d depthwise convolution with channel multiplier or (n+1)-d convolution that treats the $v$ dimension in $\boldsymbol{V}$ as an *extra spatial dimension*. We present both implementations in Appendix C.1.

As the computations are now restricted to a local scope, the lambda convolution obtains *linear* time and memory complexities with respect to the input length[9]. The lambda convolution is readily

---

[7] (Shazeer, 2019) proposes a multi-query formulation to speed-up attention-based decoding.

[8] We refer the reader to the code for more details.

[9] FLOPs (time complexity) is not necessarily a good proxy for latency on TPUs/GPUs. Eventhough the lambda convolution has linear time/space complexities, it can be slower than than the global lambda layer in practice, especially for large convolution scope sizes. See Table 4 for an example.

usable with additional functionalities such as dilation and striding and enjoys optimized implementations on specialized hardware accelerators (Nickolls & Dally, 2010; Jouppi et al., 2017). This is in stark contrast to implementations of local self-attention that require materializing feature patches of overlapping query and context blocks (Parmar et al., 2018; Ramachandran et al., 2019), increasing memory consumption and latency (see Table 4).

## 4 RELATED WORK

Table 2 reviews alternatives for capturing long-range interactions and contrasts them with the proposed multi-query lambda layer. We discuss related works in details in the Appendix D.

**Channel and linear attention** The lambda abstraction, i.e. transforming available contexts into linear functions that are applied to queries, is quite general and therefore encompasses many previous works. Closest to our work are channel and linear attention mechanisms (Hu et al., 2018c; Katharopoulos et al., 2020; Choromanski et al., 2020). Such mechanisms also capture long-range interactions without materializing attention maps and can be viewed as specific instances of a *content-only* lambda layer. Lambda layers formalize and extend such approaches to consider both content-based and *position-based* interactions, enabling their use as a stand-alone layer on highly structured data such as images. Rather than attempting to closely approximate an attention kernel as is the case with linear attention, we focus on the efficient design of contextual lambda functions and repurpose a multi-query formulation (Shazeer, 2019) to further reduce computational costs.

**Self-attention in the visual domain** In contrast to natural language processing tasks where it is now the de-facto standard, self-attention has enjoyed steady but slower adoption in the visual domain (Wang et al., 2018; Bello et al., 2019; Ramachandran et al., 2019; Carion et al., 2020). Concurrently to this work, Dosovitskiy et al. (2020) achieve a strong 88.6% accuracy on ImageNet by pre-training a Transformer on sequences of image patches on a large-scale dataset of 300M images.

## 5 EXPERIMENTS

In subsequent experiments, we evaluate lambda layers on standard computer vision benchmarks: ImageNet classification (Deng et al., 2009), COCO object detection and instance segmentation (Lin et al., 2014). The visual domain is well-suited to showcase the flexibility of lambda layers since **(1)** the memory footprint of self-attention becomes problematic for high-resolution imagery and **(2)** images are highly structured, making position-based interactions crucial.

**LambdaResNets** We construct LambdaResNets by replacing the 3x3 convolutions in the bottleneck blocks of the ResNet architecture (He et al., 2016). When replacing all such convolutions, we simply denote the name of the layer being tested (e.g. conv + channel attention or lambda layer). We denote LambdaResNets the family of *hybrid* architectures described in Table 19 (Appendix F.2). Unless specified otherwise, all lambda layers use $|k|$=16, $|h|$=4 with a scope size of $|m|$=23x23 and are implemented as in Figure 3. Additional experiments and details can be found in the Appendix.

### 5.1 LAMBDA LAYERS OUTPERFORM CONVOLUTIONS AND ATTENTION LAYERS

We first consider the standard ResNet-50 architecture with input image size 224x224. In Table 3, we compare the lambda layer against **(a)** the standard convolution (i.e. the baseline ResNet-50) **(b)** channel attention (squeeze-and-excitation) and **(c)** multiple self-attention variants. The lambda layer strongly outperforms all baselines at a fraction of the parameter cost and notably obtains a +0.8% improvement over channel attention.

### 5.2 COMPUTATIONAL BENEFITS OF LAMBDA LAYERS OVER SELF-ATTENTION

In Table 4, we compare lambda layers against self-attention and present throughputs, memory complexities and ImageNet accuracies. Our results highlight the weaknesses of self-attention: self-attention cannot model global interactions due to large memory costs, axial self-attention is still memory expensive and local self-attention is prohibitively slow. In contrast, the lambda layer can capture global interactions on high-resolution images and obtains a $+1.0\%$ improvement over local self-attention while being almost 3x faster[10]. Additionally, positional embeddings can be shared

---

[10]Latencies for local self-attention were provided privately by Ramachandran et al. (2019) based on an implementation that relies on query blocks and overlapping memory blocks (Parmar et al., 2018). Specialized attention kernels may greatly speed up local self-attention, making it a promising avenue for future research.

| Layer | Params (M) | top-1 |
|---|---|---|
| Conv (He et al., 2016)[†] | 25.6 | 76.9 |
| Conv + channel attention (Hu et al., 2018c)[†] | 28.1 | 77.6 (+0.7) |
| Conv + linear attention (Chen et al., 2018) | 33.0 | 77.0 |
| Conv + linear attention (Shen et al., 2018) | - | 77.3 (+1.2) |
| Conv + relative self-attention (Bello et al., 2019) | 25.8 | 77.7 (+1.3) |
| Local relative self-attention (Ramachandran et al., 2019) | 18.0 | 77.4 (+0.5) |
| Local relative self-attention (Hu et al., 2019) | 23.3 | 77.3 (+1.0) |
| Local relative self-attention (Zhao et al., 2020) | 20.5 | 78.2 (+1.3) |
| Lambda layer | **15.0** | **78.4** (**+1.5**) |
| Lambda layer ($|u|$=4) | **16.0** | **78.9** (**+2.0**) |

Table 3: **Comparison of the lambda layer and attention mechanisms on ImageNet classification with a ResNet50 architecture.** The lambda layer strongly outperforms attention alternatives at a fraction of the parameter cost. All models are trained in mostly similar setups (see Appendix F.3) and we include the reported improvements compared to the convolution baseline in parentheses. See Appendix C.4 for a description of the $|u|$ hyperparameter. [†] Our implementation.

across lambda layers to further reduce memory requirements, at a minimal degradation cost. Finally, the lambda convolution has linear memory complexity, which becomes practical for very large images as seen in detection or segmentation. We also find that the lambda layer outperforms local self-attention when controlling for the scope size[11] (78.1% vs 77.4% for $|m|$=7x7), suggesting that the benefits of the lambda layer go beyond improved speed and scalability.

| Layer | Space Complexity | Memory (GB) | Throughput | top-1 |
|---|---|---|---|---|
| Global self-attention | $\Theta(blhn^2)$ | 120 | OOM | OOM |
| Axial self-attention | $\Theta(blhn\sqrt{n})$ | 4.8 | 960 ex/s | 77.5 |
| Local self-attention (7x7) | $\Theta(blhnm)$ | - | 440 ex/s | 77.4 |
| Lambda layer | $\Theta(lkn^2)$ | 1.9 | 1160ex/s | **78.4** |
| Lambda layer ($|k|$=8) | $\Theta(lkn^2)$ | 0.95 | **1640** ex/s | 77.9 |
| Lambda layer (shared embeddings) | $\Theta(kn^2)$ | **0.63** | 1210 ex/s | 78.0 |
| Lambda convolution (7x7) | $\Theta(lknm)$ | - | 1100 ex/s | 78.1 |

Table 4: **The lambda layer reaches higher ImageNet accuracies while being faster and more memory-efficient than self-attention alternatives.** Memory is reported assuming full precision for a batch of 128 inputs using default hyperparameters. The memory cost for storing the lambdas matches the memory cost of activations in the rest of the network and is therefore ignored. $b$: batch size, $h$: number of heads/queries, $n$: input length, $m$: context length, $k$: query/key depth, $l$: number of layers.

## 5.3 Hybrids improve the speed-accuracy tradeoff of image classification

**Studying hybrid architectures.** In spite of the memory savings compared to self-attention, capturing global contexts with the lambda layer still incurs a quadratic time complexity (Table 2), which remains costly at high resolution. In Appendix 5.3, we study hybrid designs that use standard convolutions to capture local contexts and lambda layers to capture global contexts. We find that such convolution-lambda hybrids have increased representational power at a negligible decrease in throughput compared to their purely convolutional counterparts.

**LambdaResNets significantly improve the speed-accuracy tradeoff of ImageNet classification.** We design a family of hybrids based on our study of hybrid architectures and the scaling/training strategies from Bello et al. (2021) (Section F.2). Figure 4 presents the speed-accuracy Pareto curve of LambdaResNets compared to EfficientNets (Tan & Le, 2019) on TPUv3 hardware. In order to isolate the benefits of lambda layers, we additionally compare against the same architectures when replacing lambda layers by **(1)** standard 3x3 convolutions (denoted ResNet-RS wo/ SE) and **(2)** 3x3

---

[11]Note that the content-based lambda still captures global interactions.

convolutions with squeeze-and-excitation (denoted ResNet-RS w/ SE). All architectures are trained for 350 epochs using *the same regularization methods and evaluated at the same resolution they are trained at*. LambdaResNets outperform the baselines across all scales on the speed-accuracy trade-off.

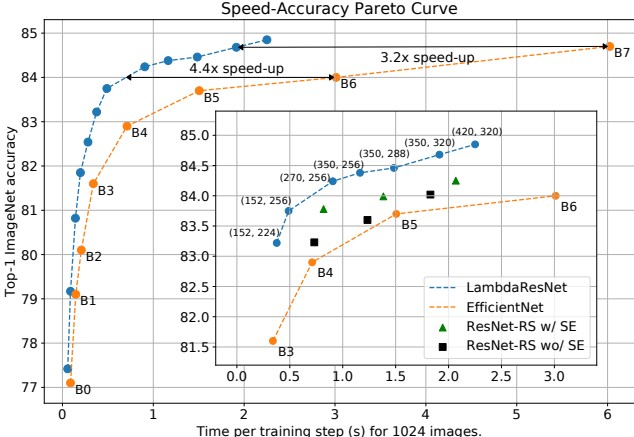

Figure 4: **Speed-accuracy comparison between LambdaResNets and EfficientNets** with matching training and regularization setups. LambdaResNets (annotated with (depth, image size)) are 3.2-4.4x faster than EfficientNets and 1.6-2.3x faster than ResNet-RS with squeeze-and-excitation, thus significantly improving the speed-accuracy Pareto curve of image classification. LambdaResNet-420 (image size 320), reaches a strong 84.9% top-1 accuracy, 0.9% over the corresponding architecture with standard 3x3 convolutions and 0.65% over the corresponding architecture with squeeze-and-excitation.

**Scaling to larger datasets with pseudo-labels**   We train LambdaResNets in a semi-supervised learning setting using 130M pseudo-labeled images from the JFT dataset, as done for training the EfficientNet-NoisyStudent checkpoints (Xie et al., 2020). Table 5 compares the throughputs and ImageNet accuracies of a representative set of models with similar accuracies when trained using the JFT dataset. LambdaResNet-152, trained and evaluated at image size 288, achieves a strong 86.7% top-1 ImageNet accuracy while being more parameter-efficient and **9.5x** faster than the EfficientNet-NoisyStudent checkpoint with the same accuracy.

| Architecture | Params (M) | Train (ex/s) | Infer (ex/s) | ImageNet top-1 |
|---|---|---|---|---|
| LambdaResNet-152 | **51** | **1620** | **6100** | 86.7 |
| EfficientNet-B7 | 66 | 170 (9.5x) | 980 (6.2x) | 86.7 |
| ViT-L/16 | 307 | 180 (9.0x) | 640 (9.5x) | **87.1** |

Table 5: **Comparison of models trained on extra data**. ViT-L/16 is pre-trained on JFT and fine-tuned on ImageNet at resolution 384x384, while EfficientNet and LambdaResNet are co-trained on ImageNet and JFT pseudo-labels. Training and inference throughput is shown for 8 TPUv3 cores.

## 6   CONCLUSION

We propose a new class of layers, termed lambda layers, which provide a scalable framework for capturing structured interactions between inputs and their contexts. Lambda layers summarize available contexts into fixed-size linear functions, termed lambdas, that are directly applied to their associated queries. The resulting neural networks, LambdaNetworks, are computationally efficient and capture long-range dependencies at a small memory cost, enabling their application to large structured inputs such as high-resolution images. Extensive experiments on computer vision tasks showcase their versatility and superiority over convolutional and attentional networks. We introduce LambdaResNets, a family of hybrid LambdaNetworks which reach excellent ImageNet accuracies and achieve up to 9.5x speed-ups over the popular EfficientNets and Vision Transformers, significantly improving the speed-accuracy tradeoff of image classification models.

ACKNOWLEDGMENTS

The author would like to thank Barret Zoph and William Fedus for endless discussions, fruitful suggestions and careful revisions; Jonathon Shlens, Mike Mozer, Prajit Ramachandran, Ashish Vaswani, Quoc Le, Neil Housby, Jakob Uszkoreit, Margaret Li, Krzysztof Choromanski for many insightful comments; Hedvig Rausing for the antarctic infographics; Zolan Brinnes for the OST; Andrew Brock, Sheng Li for assistance with profiling EfficientNets; Adam Kraft, Thang Luong and Hieu Pham for assistance with the semi-supervised experiments and the Google Brain team for useful discussions on the paper.

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

CONTENTS

# A DISCUSSION

## A.1 GENERAL DISCUSSION

**How do lambda layers compare to the attention operation?** Lambda layers scale favorably compared to self-attention. Vanilla Transformers using self-attention have $\Theta(blhn^2)$ memory footprint, whereas LambdaNetworks have $\Theta(lkn^2)$ memory footprint (or $\Theta(kn^2)$ when sharing positional embeddings across layers). This enables the use of lambda layers at higher-resolution and on larger batch sizes. Additionally, the lambda convolution enjoys a simpler and faster implementation than its local self-attention counterpart. Finally, our ImageNet experiments show that lambda layers outperforms self-attention, demonstrating that the benefits of lambda layers go beyond improved speed and scalability.

**How are lambda layers different than linear attention mechanisms?** Lambda layers generalize and extend linear attention formulations to capture *position-based* interactions, which is crucial for modeling highly structured inputs such as images (see Table 9 in Appendix E.1). As the aim is not to approximate an attention kernel, lambda layers allow for more flexible non-linearities and normalizations which we also find beneficial (see Table 11 in Appendix E.1). Finally, we propose multi-query lambda layers as a means to reduce complexity compared to the multi-head (or single-head) formulation typically used in linear attention works. Appendix D.3 presents a detailed discussion of linear attention.

**How to best use lambda layers in the visual domain?** The improved scalability, speed and ease of implementation of lambda layers compared to global or local attention makes them a strong candidate for use in the visual domain. Our ablations demonstrate that lambda layers are most beneficial in the intermediate and low-resolution stages of vision architectures when optimizing for the speed-accuracy tradeoff. It is also possible to design architectures that rely exclusively on lambda layers which can be more parameter and flops efficient. We discuss practical modeling recommendations in Appendix B.

## A.2 EXTENDING LAMBDA LAYERS TO OTHER MODALITIES

While this work focuses on static image recognition, we note that lambda layers may be instantiated to model structured interactions on structures as diverse as graphs, time series, spatial lattices, etc. We anticipate that lambda layers will be helpful in more modalities, including multimodal tasks. We discuss masked contexts and auto-regressive tasks in the Appendix C.2.

Lambda layers can be instantiated on other tasks simply by adapting the choice of structural/position embeddings to the task of interest and following the pseudo-code presented in Figure 3. The choice of embeddings dictates the memory costs of the lambda layer. The assumption underlying the $\Theta(knm)$ space complexity of the lambda layer (Section 3.2) is that *all examples in the batch share the same structure*, i.e. relative position embeddings have shape $k \times n \times m$. This assumption does not hold when the data structure is different across examples (e.g. graphs with variable edge relations between nodes), in which case embeddings have shape $b \times k \times n \times m$. In such cases, the lambda layer has $\Theta(bknm)$ space complexity, similar to self-attention.

# B  PRACTICAL MODELING RECOMMENDATIONS

**I want to make it faster on TPUs/GPUs...**    Hybrid models reach a better speed-accuracy tradeoff. Global contexts can be computationally wasteful, especially in the early high resolution layers where features lack semantic information, and can be replaced by lambda convolutions with smaller scopes (e.g. $|m|$=5x5 or 7x7) or the standard 3x3 convolution. Additionally, using a hybrid can require less tuning when starting from a working model/training setup.

**I want to make to minimize FLOPS (e.g. embedded applications)...**    Consider a hybrid with inverted bottlenecks, as done in Section E.4.2. To further reduce FLOPS, prefer lambda convolutions with smaller scopes (e.g. $|m|$=5x5 or 7x7).

**I encounter memory issues...**    Memory footprint can be reduced by sharing position embeddings across layers (especially layers with the highest resolution). Using the lambda convolution is more memory efficient. Reducing the query depth $|k|$ or increasing the number of heads $|h|$ also decreases memory consumption.

**I'm experiencing instability...**    We found it important to initialize the $\gamma$ parameter in the *last* batchnorm layer of the ResNet's bottleneck blocks to 0 (this is the default in most codebases). Normalizing the keys (i.e. with the softmax) along the context's length is important. Early experiments which employed 2 lambda layers sequentially in the same residual block were unstable, suggesting that using 2 lambda layers in sequence should be avoided.

**Which implementation of the lambda convolution should I use?**    In our experiments using Tensorflow 1.x on TPUv3 hardware, we found both the n-d depthwise and (n+1)-d convolution implementations to have similar speed. We point out that this can vary across software/hardware stacks.

**What if my task doesn't require position-based interactions?**    Computational costs in the lambda layer are dominated by position-based interactions. If your task doesn't require them, you can try the content-only lambda layer or any other linear attention mechanism. We recommend using the *multi-query* formulation (as opposed to the usual multi-head) and scaling other dimensions of the model.

## C    ADDITIONAL VARIANTS

### C.1    COMPLETE CODE WITH LAMBDA CONVOLUTION

```
# b: batch, n: input length, m: context length, r: scope size,
# k: query/key depth, v: value depth, h: number of heads, d: output dimension.
def compute_position_lambdas(embeddings, values, impl='einsum'):
    if impl == 'einsum': # embeddings shape: [n, m, k]
        position_lambdas = einsum(embeddings, values, 'nmk,bmv->bnkv')
    else: # embeddings shape: [r, k]
        if impl == 'conv':
            embeddings = reshape(embeddings, [r, 1, 1, k])
            values = reshape(values, [b, n, v, 1])
            position_lambdas = conv2d(values, embeddings)
        elif impl == 'depthwise_conv':
            # Reshape and tile embeddings to [r, v, k] shape
            embeddings = reshape(embeddings, [r, 1, k])
            embeddings = tile(embeddings, [1, v, 1])
            position_lambdas = depthwise_conv1d(values, embeddings)
        # Transpose from shape [b, n, v, k] to shape [b, n, k, v]
        position_lambdas = transpose(position_lambdas, [0, 1, 3, 2])
    return position_lambdas

def lambda_layer(queries, keys, embeddings, values, impl='einsum'):
    """Multi-query lambda layer."""
    content_lambda = einsum(softmax(keys), values, 'bmk,bmv->bkv')
    position_lambdas = compute_position_lambdas(embeddings, values, impl=impl)
    content_output = einsum(queries, content_lambda, 'bhnk,bkv->bnhv')
    position_output = einsum(queries, position_lambdas, 'bhnk,bnkv->bnhv')
    output = reshape(content_output + position_output, [b, n, d])
    return output
```

Figure 5: **Pseudo-code for the multi-query lambda layer and the 1d lambda convolution.** A n-d lambda convolution can equivalently be implemented via a regular (n+1)-d convolution or a n-d depthwise convolution with channel multiplier. The embeddings can be made to satisfy various conditions (e.g. translation equivariance and masking) when computing positional lambdas with the einsum implementation.

### C.2    GENERATING LAMBDAS FROM MASKED CONTEXTS

In some applications, such as denoising tasks or auto-regressive training, it is necessary to restrict interactions to a sub-context $\mathcal{C}_n \subset \mathcal{C}$ when generating $\boldsymbol{\lambda}_n$ for query position $n$. For example, *parallel* auto-regressive training requires masking the future to ensure that the output $\boldsymbol{y}_n$ only depends on past context positions $m < n$. Self-attention achieves this by zeroing out the irrelevant attention weights $\boldsymbol{a}_{nm'} = 0 \; \forall m' \notin \mathcal{C}_n$, thus guaranteeing that $\boldsymbol{y}_n = \sum_m \boldsymbol{a}_{nm} \boldsymbol{v}_m$ only depends on $\mathcal{C}_n$.

Similarly, one can block interactions between queries and masked context positions when generating lambdas by applying a mask before summing the contributions of context positions. As long as the mask is shared across all elements in the batch, computing masked lambdas does not require materializing per-example attention maps and the complexities are the same as for global context case. See Figure 6 for an implementation.

### C.3    MULTI-HEAD VS MULTI-QUERY LAMBDA LAYERS

In this section, we motivate using a multi-query formulation as opposed to the usual multi-head formulation used in self-attention. Figure 7 presents the implementation of a multi-head lambda layer. Table 6 compares complexities for multi-head and multi-query lambda layers. Using a multi-query formulation reduces computations by a factor of $|h|$ (the number of queries per lambda) compared to the multi-head formulation. We also found in early experimentation that multi-query lambdas yield a better speed-accuracy trade-off. Additionally, the multi-head lambda layer does not enjoy a simple local implementation as the lambda convolution.

```
def masked_lambda_layer(queries, normalized_keys, embeddings, values, mask):
    """Masked multi-query lambda layer.
    Args:
      queries: a tensor with shape [b, h, n, k].
      normalized_keys: a tensor with shape [b, m, k].
      embeddings: a tensor with shape [k, n, m].
      values: a tensor with shape [b, m, v].
      mask: a tensor of 0 and 1s with shape [n, m].
    """
    # We show the general case but a cumulative sum may be faster for masking the future.
    # Note that each query now also has its own content_lambda since every query
    # interacts with a different context.
    # Keys should be normalized by only considering the elements in their contexts.
    content_mu = einsum(normalized_keys, values, 'bmk,bmv->bmkv')
    content_lambdas = einsum(content_mu, mask, 'bmkv,nm->bnkv')
    embeddings = einsum(embeddings, mask, 'knm,nm->knm') # apply mask to embeddings
    position_lambdas = einsum(embeddings, values, 'knm,bmv->bnkv')
    content_output = einsum(queries, content_lambda, 'bhnk,bnkv->bnhv')
    position_output = einsum(queries, position_lambdas, 'bhnk,bnkv->bnhv')
    output = reshape(content_output + position_output, [b, n, d])
    return output
```

Figure 6: **Pseudo-code for *masked* multi-query lambda layer.**

```
def multihead_lambda_layer(queries, keys, embeddings, values, impl='einsum'):
    """Multi-head lambda layer."""
    content_lambda = einsum(softmax(keys), values, 'bhmk,bhmv->bhkv')
    position_lambdas = einsum(embeddings, values, 'hnmk,bhmv->bnhkv')
    content_output = einsum(queries, content_lambda, 'bhnk,bhkv->bnhv')
    position_output = einsum(queries, position_lambdas, 'bhnk,bnkv->bnhv')
    output = reshape(content_output + position_output, [b, n, d])
    return output
```

Figure 7: **Pseudo-code for the *multi-head* lambda layer.** This is only shown as an example as we recommend the *multi-query* lambda laayer instead.

| Operation | Time complexity | Space complexity |
|---|---|---|
| Multi-head lambda layer | $\Theta(bnmkd)$ | $\Theta(knm + bnkd)$ |
| Multi-query lambda layer | $\Theta(bnmkd/h)$ | $\Theta(hknm + bnkd/h)$ |

Table 6: **Complexity comparison between a multi-head and a multi-query lambda layer.** Using a multi-query formulation reduces complexity by a factor $|h|$ (the number of queries per lambda) compared to the standard multi-head formulation.

### C.4 ADDING EXPRESSIVITY WITH AN EXTRA DIMENSION

We briefly experiment with a variant that enables increasing the cost of *computing* the lambdas while keeping the cost of *applying* them constant. This is achieved by introducing an additional dimension, termed the intra-depth with corresponding hyperparameter $|u|$, in keys, position embeddings and values. Each key (or positional embedding) is now a $|k| \times |u|$ matrix instead of a $|k|$-dimensional vector. Similarly, each value is now a $|v| \times |u|$ matrix instead of a $|v|$-dimensional vector. The lambdas are obtained via summing over context positions *and the intra-depth position* $|u|$ and have $|k| \times |v|$ shape similar to the default case. See Figure 8 for an implementation and Table 7 for the complexities. Experiments (see Appendix E.1) demonstrate that this variant results in accuracy improvements but we find that using $|u|$=1 (i.e. the default case) is optimal when controlling for speed on modern machine learning accelerators.

```
def compute_position_lambdas(embeddings, values, impl='einsum'):
    """Compute position lambdas with intra-depth u."""
    if impl == 'conv':
        # values: [b, n, v, u] shape
        # embeddings: [r, 1, u, k] shape
        position_lambdas = conv2d(values, embeddings)
        # Transpose from shape [b, n, v, k] to shape [b, n, k, v]
        position_lambdas = transpose(position_lambdas, [0, 1, 3, 2])
    elif impl == 'einsum':
        # embeddings: [k, n, m, u] shape
        position_lambdas = einsum(embeddings, values, 'knmu,bmvu->bnkv')
    return position_lambdas

def lambda_layer(queries, keys, embeddings, values, impl='einsum'):
    """Multi-query lambda layer with intra-depth u."""
    content_lambda = einsum(softmax(keys), values, 'bmku,bmvu->bkv')
    position_lambdas = compute_position_lambdas(embeddings, values, lambda_conv)
    content_output = einsum(queries, content_lambda, 'bhnk,bkv->bnhv')
    position_output = einsum(queries, position_lambdas, 'bhnk,bnkv->bnhv')
    output = reshape(content_output + position_output, [b, n, d])
    return output
```

Figure 8: **Pseudo-code for the multi-query lambda layer with intra-depth** $|u|$. Lambdas are obtained by reducing over the context positions and the intra-depth dimension. This variant allocates more computation for generating the lambdas while keeping the cost of applying them constant. The equivalent n-d lambda convolution can be implemented with a regular (n+1)-d convolution.

| Operation | Time complexity | Space complexity |
|---|---|---|
| Lambda layer ($|u| > 1$) | $\Theta(bnmkud/h)$ | $\Theta(knmu + bnkv)$ |

Table 7: **Complexity for a multi-query lambda layer with intra-depth** $|u|$.

# D  ADDITIONAL RELATED WORK

In this section, we review the attention operation and related works on improving its scalability. We discuss connections between lambda layers and channel, spatial or linear attention mechanisms and show how they can be cast as *less flexible specific instances* of lambda layers. We conclude with a brief review of self-attention in the visual domain and discuss connections with expert models.

## D.1  SOFTMAX ATTENTION

**Softmax attention**  Softmax-attention produces a distribution over the context for each query $q_n$ as $a_n = \text{softmax}(Kq_n) \in \mathbb{R}^{|m|}$ where the keys $K$ are obtained from the context $C$. The attention distribution $a_n$ is then used to form a linear combination of values $V$ obtained from the context as $y_n = V^T a_n = \sum_m a_{nm} v_m \in \mathbb{R}^{|v|}$. As we take a weighted sum of the values[12], we transform the query $q_n$ into the output $y_n$ and discard its attention distribution $a_n$. This operation captures content-based interactions, but not position-based interactions.

**Relative attention**  In order to model position-based interactions, relative attention (Shaw et al., 2018) introduces a learned matrix of $|m|$ positional embeddings $E_n \in \mathbb{R}^{|m| \times |k|}$ and computes the attention distribution as $a_n = \text{softmax}((K + E_n)q_n) \in \mathbb{R}^{|m|}$. The attention distribution now also depends on the query position $n$ relative to positions of context elements $m$. Relative attention therefore captures both content-based and position-based interactions.

---

[12]Sometimes the attention operation is instead used to *point* to specific context elements (Vinyals et al., 2015; Bello et al., 2016), which is not supported by lambda layers.

## D.2 Sparse attention

A significant challenge in applying (relative) attention to large inputs comes from the *quadratic* $\Theta(|bnm|)$ memory footprint required to store attention maps. Many recent works therefore propose to impose specific patterns to the attention maps as a means to reduce the context size $|m|$ and consequently the memory footprint of the attention operation. These approaches include:

- *local* attention patterns (Dai et al., 2019; Parmar et al., 2018; Ramachandran et al., 2019)
- *axial* attention patterns (Ho et al., 2019; Wang et al., 2020a; Shen et al., 2020)
- *static sparse* attention patterns (Child et al.; Beltagy et al., 2020)
- *dynamic sparse* attention patterns (Kitaev et al., 2020)

See Tay et al. (2020) for a review. Their implementations can be rather complex, sometimes require low-level kernel implementations to get computational benefits or may rely on specific assumptions on the shape of the inputs (e.g., axial attention). In contrast, lambda layers are simple to implement for both global and local contexts using simple einsum and convolution primitives and capture *dense* content and *position-based* interactions with no assumptions on the input shape.

## D.3 Linear attention: connections and differences

Another approach to reduce computational requirements of attention mechanisms consists in approximating the attention operation in linear space and time complexity, which is referred to as linear (or efficient) attention. Linear attention mechanisms date back to de Brébisson & Vincent (2016); Britz et al. (2017) and were later introduced in the visual domain by Chen et al. (2018); Shen et al. (2018). They are recently enjoying a resurgence of popularity with many works modifying the popular Transformer architecture for sequential processing applications (Katharopoulos et al., 2020; Wang et al., 2020b; Choromanski et al., 2020; Xiong et al., 2021).

**Linear attention via kernel factorization**   Linear attention is typically obtained by reinterpreting attention as a similarity kernel and leveraging a low-rank kernel factorization as

$$\text{Attention}(\boldsymbol{Q}, \boldsymbol{K}, \boldsymbol{V}) = \text{softmax}(\boldsymbol{Q}\boldsymbol{K}^T)\boldsymbol{V} \sim \phi(\boldsymbol{Q})(\phi(\boldsymbol{K}^T)\boldsymbol{V}) \tag{3}$$

for some feature function $\phi$. Computing $\phi(\boldsymbol{K}^T)\boldsymbol{V} \in \mathbb{R}^{|k| \times |v|}$ first bypasses the need to materialize the attention maps $\phi(\boldsymbol{Q})\phi(\boldsymbol{K}^T)$ and the operation therefore has *linear* complexity with respect to the input length $|n|$.

Multiple choices for the feature function $\phi$ have been proposed. For example, Katharopoulos et al. (2020) use $\phi(\boldsymbol{x}) = \text{elu}(\boldsymbol{x}) + 1$, while Choromanski et al. (2020) use positive orthogonal random features to approximate the original softmax attention kernel. In the visual domain, both Chen et al. (2018) and Shen et al. (2018) use $\phi(\boldsymbol{x}) = \text{softmax}(\boldsymbol{x})$. This choice is made to guarantee that the rows of the (non-materialized) attention maps $\phi(\boldsymbol{Q})\phi(\boldsymbol{K})^T$ sum to 1 as is the case in the regular attention operation.

We discuss the main differences between lambda layers and linear attention mechanisms.

**1) Lambda layers extend linear attention to also consider position-based interactions.**   The kernel approximation from Equation 3 can be rewritten for a single query $\boldsymbol{q}_n$ as

$$\boldsymbol{y}_n = (\phi(\boldsymbol{K})^T\boldsymbol{V})^T\phi(\boldsymbol{q}_n) \tag{4}$$

which resembles the output of the *content lambda* $\boldsymbol{y}_n^c = (\boldsymbol{\lambda}^c)^T\boldsymbol{q}_n = (\bar{\boldsymbol{K}}^T\boldsymbol{V})^T\boldsymbol{q}_n$ from Equation 1. Lambda layers extend linear attention mechanisms to also consider position-based interactions as

$$\boldsymbol{y}_n = \boldsymbol{\lambda}_n^T\boldsymbol{q}_n = (\boldsymbol{\lambda}^c + \boldsymbol{\lambda}_n^p)^T\boldsymbol{q}_n = ((\bar{\boldsymbol{K}} + \boldsymbol{E}_n)^T\boldsymbol{V})^T\boldsymbol{q}_n \tag{5}$$

In the above equation, computing the position (or content) lambda has $\Theta(bmkv)$ time complexity. As the position lambdas are not shared across query positions $n$, this cost is repeated for all $|n|$ queries, leading to a total time complexity $\Theta(bnmkv)$. Unlike linear attention mechanisms, lambda layers have *quadratic time complexity* with respect to the input length (in the global context case) because they consider position-based interactions.

**2) Lambda layers do not necessarily attempt to approximate an attention kernel.** While approximations of the attention kernel are theoretically motivated, we argue that they may be unnecessarily restrictive. For example, the kernel approximation in Equation 3 requires the *same* feature function $\phi$ on both $Q$ and $K$ and precludes the use of more flexible non-linearities and normalization schemes. In contrast, lambda layers do not attempt to approximate an attention kernel. This simplifies their design and allows for more flexible non-linearity and normalization schemes, which we find useful in our ablations (See Table 11 in Appendix E.1). Considering the position embeddings independently of the keys notably enables a simple and efficient local implementation with the lambda convolution. Approximating the *relative* attention kernel would require normalizing the position embeddings with the keys (i.e., $\phi(K + E_n)$ instead of $\phi(K) + E_n$), which cannot be implemented in the local context case with a convolution.

**3) The lambda abstraction reveals the computational benefits of the multi-query formulation.** Finally, this work proposes to abstract the $\bar{K}^T V$ and $E_n^T V$ matrices as linear functions (the *content* and *position* lambdas) that are directly applied to the queries. The lambda abstraction reveals the benefits of multi-query formulation (as opposed to the traditional multi-head attention formulation) as a means to reduce computational costs.

### D.4   CASTING CHANNEL AND SPATIAL ATTENTION AS LAMBDA LAYERS.

We show that the lambda abstraction generalizes *channel* and *spatial* attention mechanisms, both of which can be viewed as specific instances of lambda layers. This observation is consistent with our experiments which demonstrate that lambda layers outperform both channel and spatial attention while being more computationally efficient.

**Channel attention**   *Channel attention* mechanisms, such as Squeeze-and-Excitation (SE) (Hu et al., 2018c;b) and FiLM layers (Perez et al., 2017), recalibrate features via cross-channel interactions by aggregating signals from the entire feature map. In particular, the SE operation can be written as $y_{nk} = w_k q_{nk}$ where $w_k$ is the excitation weight for channel $k$ in the query $q_n$. This can be viewed as using a *diagonal* lambda which is *shared across query positions* $\lambda_n = diag(w_1 \cdots w_{|k|})$. Channel attention mechanisms have proven useful to complement convolutions but cannot be used as a stand-alone layer as they discard spatial information.

**Spatial attention**   Conversely, *spatial attention* mechanisms, reweigh each position based on signals aggregated from all channels (Xu et al., 2015; Park et al., 2018; Woo et al., 2018). These mechanisms can be written as $y_{nk} = w_n q_{nk}$ where $w_n$ is the attention weight for position $n$ in the input query $Q$. This can be viewed as using (position-dependent) scalar lambdas $\lambda_n = w_n \mathbb{I}$ where $\mathbb{I}$ is the identity matrix. Spatial attention has also proven helpful to complement convolutions but cannot be used as a stand-alone layer as it discards channel information.

### D.5   SELF-ATTENTION IN THE VISUAL DOMAIN

Self-attention has been used in a myriad of tasks in the visual domain. These include image classification (Bello et al., 2019; Ramachandran et al., 2019; Cordonnier et al., 2019; Zhao et al., 2020; Wu et al., 2020; Dosovitskiy et al., 2020); object detection and object-centric tasks (Wang et al., 2018; Hu et al., 2018a; Carion et al., 2020; Locatello et al., 2020); video tasks (Sun et al., 2019; Liao et al., 2019); autoregressive/adversarial generative modeling (Parmar et al., 2018; Zhang et al., 2019; Brock et al., 2019; Chen et al., 2020a) and multi-modal text-vision tasks (Chen et al., 2020b; Lu et al., 2019; Li et al., 2019; Radford et al., 2021)

The first use of self-attention in vision dates back to the non-local block (Wang et al., 2018), which added a single-head global self-attention residual in the low resolution stages of a ConvNet for long-range dependency modeling. The non-local block has proven useful to complement convolutions but cannot be used as a stand-alone layer as it does not model position-based interactions.

***Global* relative attention replaces convolutions at *low* resolution.**   Bello et al. (2019) introduced a 2d relative attention mechanism that proved competitive as a replacement to convolutions but gives even stronger results when used to concatenate convolutional features with self-attention features. The spatial convolutions in the bottleneck block of the ResNet architecture were replaced with a

*global multi-head* self-attention mechanism with *2d relative position embeddings*. Due to the large memory constraints of global attention, this operation was restricted to low resolution feature maps and the proposed architecture was a *conv-transformer* hybrid.

A similar hybrid design has recently been revisited by Srinivas et al. (2021) using modern training and scaling techniques. Srinivas et al. (2021), rather than concatenating convolutional feature maps, propose to use a stride of 1 in the last stage of the ResNet architecture for improved performance.

***Local/axial* relative attention replaces convolutions at *high* resolution.** The large memory footprint of global attention was quickly solved by multiple works which proposed to limit the size of the attention contexts such as *local* attention (Ramachandran et al., 2019; Hu et al., 2019) and *axial* attention (Ho et al., 2019; Wang et al., 2020a; Shen et al., 2020) (See Section D.2). Such approaches enable using attention at higher resolution and facilitate fully-attentional models but can be slow due to the use of specialized attention patterns.

**Scaling trumps inductive bias** Concurrently to this work, ViT (Dosovitskiy et al., 2020) propose to simply apply attention on *pixel patches* (as opposed to individual pixels) as a remedy to large memory requirements. While patch-based attention does not maintain accurate positional information or translation equivariance, the loss of inductive bias is recovered by pre-training on large-scale datasets (e.g. 300M images). Most remarkably, ViT achieves close to state-of-the-art accuracy when fine-tuned on the ImageNet dataset, while requiring less training compute that convolutional alternatives (Kolesnikov et al., 2020; Xie et al., 2020). This result has reinvigorated interest in using self-attention in the visual domain with multiple follow-up works already building upon this approach (Touvron et al., 2021)[13]. In spite of the impressive image classification results, concerns remain as to whether the patch-based approach can scale to larger images and transfer to tasks that require precise localization such as detection.

We stress that reducing memory by working with pixel patches is orthogonal to the specific operation used and that lambda layers (or linear attention) can also be used on pixel patches.

### D.6 HYPERNETWORKS, EXPERT MODELS AND CONTEXT-DEPENDENT WEIGHTS

LambdaNetworks generate their own computations, i.e. lambdas such that $y_n = \lambda_n^T q_n$. As such, they can alternatively be viewed as an extension of HyperNetworks (Ha et al., 2016) that dynamically generate their computations based on *structured contextual information*. The concept of generating context-dependent weights is also related to fast weights (Ba et al., 2016).

Lastly, LambdaNetworks share some connections with sparsely-activated expert models (Shazeer et al., 2017; Fedus et al., 2021). Whereas sparsely-activated expert models *select* the computation (i.e. the lambda) from a bank of weights based on the input query, LambdaNetworks *generate* their computations based on contextual information.

---

[13]Most follow-up works advertise improvements over ViT on smaller datasets which is not the intended purpose of ViT.

# E  ADDITIONAL EXPERIMENTS

## E.1  ABLATION STUDY

We perform several ablations and validate the importance of positional interactions, long-range interactions and flexible normalization schemes. Unless specified otherwise, all experimental results in this section report ImageNet accuracies obtained by training a LambdaNetwork architecture that replaces the spatial convolutions in the ResNet-50 with lambda layers.

**Varying query depth, number of heads and intra-depth.**  Table 8 presents the impact of the query depth $|k|$, number of heads $|h|$ and intra depth $|u|$ on performance (See Appendix C.4 for a presentation of the intra-depth $|u|$). Our experiments indicate that the lambda layer outperforms convolutional and attentional baselines for a wide range of hyperparameters, demonstrating the robustness of the method.

| $|k|$ | $|h|$ | $|u|$ | Params (M) | top-1 |
|---|---|---|---|---|
| ResNet baseline | | | 25.6 | 76.9 |
| 8 | 2 | 1 | 14.8 | 77.2 |
| 8 | 16 | 1 | 15.6 | 77.9 |
| 2 | 4 | 1 | 14.7 | 77.4 |
| 4 | 4 | 1 | 14.7 | 77.6 |
| 8 | 4 | 1 | 14.8 | 77.9 |
| 16 | 4 | 1 | 15.0 | 78.4 |
| 32 | 4 | 1 | 15.4 | 78.4 |
| 2 | 8 | 1 | 14.7 | 77.8 |
| 4 | 8 | 1 | 14.7 | 77.7 |
| 8 | 8 | 1 | 14.7 | 77.9 |
| 16 | 8 | 1 | 15.1 | 78.1 |
| 32 | 8 | 1 | 15.7 | 78.5 |
| 8 | 8 | 4 | 15.3 | 78.4 |
| 8 | 8 | 8 | 16.0 | 78.6 |
| 16 | 4 | 4 | 16.0 | 78.9 |

Table 8: **Ablations on the ImageNet classification task when using the lambda layer in a ResNet50 architecture.** All configurations outpeform the convolutional baseline at a lower parameter cost. As expected, we get additional improvements by increasing the query depth $|k|$ or intra-depth $|u|$. The number of heads is best set to intermediate values such as $|h|$=4. A large number of heads $|h|$ excessively decreases the value depth $|v| = d/|h|$, while a small number of heads translates to too few queries, both of which hurt performance.

**Content vs position interactions**  Table 9 presents the relative importance of content-based and position-based interactions on the ImageNet classification task. We find that position-based interactions are crucial to reach high accuracies, while content-based interactions only bring marginal improvements over position-based interactions[14].

| Content | Position | Params (M) | FLOPS (B) | top-1 |
|---|---|---|---|---|
| ✓ | ✗ | 14.9 | 5.0 | 68.8 |
| ✗ | ✓ | 14.9 | 11.9 | 78.1 |
| ✓ | ✓ | 14.9 | 12.0 | 78.4 |

Table 9: **Contributions of content and positional interactions**. As expected, positional interactions are crucial to perform well on the image classification task.

---

[14]This observation is challenged by concurrent work (Dosovitskiy et al., 2020) which demonstrates that content-based interactions can be sufficient for image classification when pre-training on large scale datasets (e.g. 300M images).

**Importance of scope size**    The small memory footprint of LambdaNetworks enables considering global contexts, even at relatively high resolution. Table 10 presents flops counts and top-1 ImageNet accuracies when varying scope sizes in a LambdaNetwork architecture. We find benefits from using larger scopes, with a plateau around $|m|$=15x15, which validates the importance of longer range interactions compared to the usual 3x3 spatial convolutions used in the ResNet architecture. In our main experiments, we choose $|m|$=23x23 as the default to account for experiments that use larger image sizes.

| Scope size $|m|$ | 3x3 | 7x7 | 15x15 | 23x23 | 31x31 | global |
|---|---|---|---|---|---|---|
| FLOPS (B) | 5.7 | 6.1 | 7.8 | 10.0 | 12.4 | 19.4 |
| Top-1 Accuracy | 77.6 | 78.2 | 78.5 | 78.3 | 78.5 | 78.4 |

Table 10: **Impact of varying the scope size for positional lambdas on the ImageNet classification task**. We replace the 3x3 spatial convolutions in the *last 2 stages* of a ResNet-50 with lambda layers (input image size is 224x224). Flops significantly increase with the scope size, however we stress that larger scopes do not translate to slower latencies when using the einsum implementation (see Figure 3).

**Normalization**    Table 11 ablates normalization operations in the design of the lambda layer. We find that normalizing the keys is crucial for performance and that other normalization functions besides the softmax can be considered. Applying batch normalization to the queries and values is also helpful.

| Normalization | top-1 |
|---|---|
| Softmax on keys (default) | 78.4 |
| Softmax on keys & Softmax on queries | 78.1 |
| L2 normalization on keys | 78.0 |
| No normalization on keys | 70.0 |
| No batch normalization on queries and values | 76.2 |

Table 11: **Impact of normalization schemes in the lambda layer.** Normalization of the keys along the context spatial dimension $m$, normalization of the queries along the query depth $k$.

### E.2   HYBRID MODELS STUDY

In this section, we study hybrid designs that use standard convolutions to capture local contexts and lambda layers to capture global contexts.[15]

**Where are lambda layers most useful?**    Table 12 presents the throughputs and accuracies of hybrid LambdaNetwork architectures as a function of the location of convolutions and lambda layers in a ResNet-50 architecture. We observe that lambda layers are most helpful in the last two stages (commonly referred to as *c4* and *c5*) when considering their speed-accuracy tradeoff. We refer to architectures that replaces 3x3 convolutions in the last 2 stages of the ResNet with lambda layers as LambdaResNet-**C4**.

**Further pushing the speed-accuracy Pareto frontier.**    In Table 13, we further study how throughput and accuracy are impacted by the number of lambda layers in the *c4* stage. Our results reveal that most benefits from lambda layers can be obtained by **(a)** replacing a few 3x3 convolutions with lambda layers in the *c4* stage and **(b)** replacing all 3x3 convolutions in *c5*. The resulting hybrid LambdaResNets architectures have increased representational power at a virtually negligible decrease in throughput compared to their vanilla ResNet counterparts. Table 19 presents the detailed block configurations and placement of lambda layers for our family of LambdaResNets.

---

[15]We could alternatively use the lambda convolution to capture local contexts.

| Architecture | Params (M) | Throughput | top-1 |
|---|---|---|---|
| C → C → C → C | 25.6 | 7240 ex/s | 76.9 |
| L → C → C → C | 25.5 | 1880 ex/s | 77.3 |
| L → L → C → C | 25.0 | 1280 ex/s | 77.2 |
| L → L → L → C | 21.7 | 1160 ex/s | 77.8 |
| L → L → L → L | 15.0 | 1160 ex/s | 78.4 |
| C → L → L → L | 15.1 | 2200 ex/s | 78.3 |
| C → C → L → L | 15.4 | 4980 ex/s | 78.3 |
| C → C → C → L | 18.8 | 7160 ex/s | 77.3 |

Table 12: **Hybrid models achieve a better speed-accuracy trade-off.** Inference throughput and top-1 accuracy as a function of lambda (L) vs convolution (C) layers' placement in a ResNet50 architecture on 224x224 inputs. Lambda layers in the *c5* stage incur almost no speed decrease compared to standard 3x3 convolutions. Lambda layers in the *c4* stage are relatively slower than standard 3x3 convolutions but yield significant accuracy gains.

| Config | Image size | Params (M) | Throughput | top-1 |
|---|---|---|---|---|
| ResNet-101 wo/ SE | 224 | 44.6 | 4600 ex/s | 81.3 |
| ResNet-101 w/ SE | 224 | 63.6 | 4000 ex/s | 81.8 |
| LambdaResNet-101 | 224 | 36.9 | **4040 ex/s** | **82.3** |
| LambdaResNet-101-C4 | 224 | 26.0 | 2560 ex/s | 82.6 |
| ResNet-152 wo/ SE | 256 | 60.2 | 2780 ex/s | 82.5 |
| ResNet-152 w/ SE | 256 | 86.6 | 2400 ex/s | 83.0 |
| LambdaResNet-152 | 256 | 51.4 | **2400 ex/s** | **83.4** |
| LambdaResNet-152-C4 | 256 | 35.1 | 1480 ex/s | 83.4 |

Table 13: **Impact of number of lambda layers in the c4 stage of LambdaResNets.** Most benefits from lambda layers can be obtained by having a few lambda layers in the *c4* stage. Such hybrid designs maximize the speed-accuracy tradeoff. LambdaResNet-C4 architectures exclusively employ lambda layers in *c4* and *c5*. LambdaResNet block configurations can be found in Table 19. Models are trained for 350 epochs on the ImageNet classification task.

**Comparing hybrid lambda vs attention models.** The memory savings of lambda layers compared to attention are less significant in the aforementioned hybrid design, since the operations occur at lower resolution. Therefore, it is natural to ask whether lambda layers still have benefits over self-attention when considering hybrid designs. We consider our largest hybrid as an example (see Table 19). LambdaResNet-420 is trained on 320x320 inputs, employs 8 lambda layers in *c4* and can fit 32 examples per TPU-v3 core. This adds up to a cost of 38.4MB for lambda layers (4.8MB if sharing positional embeddings), whereas using attention layers instead would incur 0.625GB. The increase might not be significant in practice and it will be interesting to carefully benchmark the hybrid attention variants[16]. We point that experiments from Table 4 suggest that the benefits of lambda layers go beyond improved scalability and stress that the memory savings are more pronounced for tasks that require larger inputs such as object detection.

### E.3 OBJECT DETECTION AND INSTANCE SEGMENTATION RESULTS

In Table 14, we evaluate LambdaResNets as a backbone in Mask-RCNN (He et al., 2017) on the COCO object detection and instance segmentation tasks. Using lambda layers yields consistent gains across all object sizes, especially the small objects which are the hardest to locate. This indicates that lambda layers are also competitive for more complex visual tasks that require localization information.

---

[16]We will benchmark such architectures in a future version of this draft.

| Backbone | $AP^{bb}_{coco}$ | $AP^{bb}_{s/m/l}$ | $AP^{mask}_{coco}$ | $AP^{mask}_{s/m/l}$ |
|---|---|---|---|---|
| ResNet-101 | 48.2 | 29.9 / 50.9 / 64.9 | 42.6 | 24.2 / 45.6 / 60.0 |
| ResNet-101 + SE | 48.5 (+0.3) | 29.9 (+0.0) / 51.5 / 65.3 | 42.8 (+0.2) | 24.0 (-0.2) / 46.0 / 60.2 |
| LambdaResNet-101 | **49.4** (+1.2) | **31.7** (+1.8) / **52.2** / **65.6** | **43.5** (+0.9) | **25.9** (+1.7) / **46.5** / **60.8** |
| ResNet-152 | 48.9 | 29.9 / 51.8 / 66.0 | 43.2 | 24.2 / 46.1 / 61.2 |
| ResNet-152 + SE | 49.4 (+0.5) | 30.0 (+0.1) / 52.3 / 66.7 | 43.5 (+0.3) | 24.6 (+0.4) / 46.8 / 61.8 |
| LambdaResNet-152 | **50.0** (+1.1) | **31.8** (+1.9) / **53.4** / **67.0** | **43.9** (+0.7) | **25.5** (+1.3) / **47.3** / **62.0** |

Table 14: **COCO object detection and instance segmentation with Mask-RCNN architecture on 1024x1024 inputs**. We compare LambdaResNets against ResNets with or without squeeze-and-excitation (SE) and report Mean Average Precision (AP) for small, medium, large objects ($AP_{s/m/l}$). Using lambda layers yields consistent gains across all object sizes, especially small objects.

### E.4 PARAMETER AND FLOPS EFFICIENCY RESULTS

#### E.4.1 COMPUTATIONAL EFFICIENCY COMPARISONS TO LARGE EFFICIENTNETS

In Table 15 and Table 16, we showcase the parameter and flops-efficiency of LambdaNetworks. We find that LambdaResNet-C4 which replaces the 3x3 convolutions in the last 2 stages of the ResNet architecture, where they incur the highest parameter costs, improves upon parameter and flops efficiency of large EfficientNets. These results are significant because EfficientNets were specifically designed by neural architecture search (Zoph & Le, 2017) to minimize computational costs using highly computationally efficient depthwise convolutions (Tan & Le, 2019).

| Architecture | Image size | Params (M) | top-1 |
|---|---|---|---|
| EfficientNet-B6 | 528x528 | 43 | 84.0 |
| LambdaResNet-152-C4 | 320x320 | **35** | 84.0 |
| LambdaResNet-200-C4 | 320x320 | 42 | **84.3** |

Table 15: **Parameter-efficiency comparison between LambdaResNet-C4 and EfficientNet-B6**. LambdaResNet-C4 is more parameter-efficient in spite of using a smaller image size. Increasing the image size would likely result in improved accuracy while keeping the number of parameters fixed. Models are trained for 350 epochs.

| Architecture | Image size | Flops (G) | top-1 |
|---|---|---|---|
| EfficientNet-B6 | 528x528 | 38 | **84.0** |
| LambdaResNet-270-C4 ($|m|$=7x7) | 256x256 | **34** | **84.0** |

Table 16: **Flops-efficiency comparison between LambdaResNet-C4 and EfficientNet-B6**. We use smaller local scopes ($|m|$=7x7) to reduce FLOPS in the lambda layers. Models are trained for 350 epochs.

#### E.4.2 LAMBDA LAYERS IN A RESOURCE CONSTRAINED SCENARIO

Lastly, we briefly study lambda layers in a resource-constrained scenario using the MobileNetv2 architecture (Sandler et al., 2018). MobileNets (Howard et al., 2017; Sandler et al., 2018; Howard et al., 2019) employ lightweight inverted bottleneck blocks which consist of the following sequence: 1) a pointwise convolution for expanding the number of channels, 2) a depthwise convolution for spatial mixing and 3) a final pointwise convolution for channel mixing. The use of a depthwise convolution (as opposed to a regular convolution) reduces parameters and flops, making inverted bottlenecks particularly well-suited for embedded applications.

**Lightweight lambda block.** We construct a lightweight lambda block as follows. We replace the depthwise convolution in the inverted bottleneck with a lambda convolution with small scope size $|m|$=5x5, query depth $|k|$=32, number of heads $|h|$=4. We also change the first pointwise

convolution to output the same number of channels (instead of increasing the number of channels) to further reduce computations.

**Adding lambda layers in MobileNetv2.**   We wish to assess whether lambda layers can improve the flops-accuracy (or parameter-accuracy) tradeoff of mobilenet architectures. We experiment with a simple strategy of replacing a few inverted bottlenecks with our proposed lightweight lambda block, so that the resulting architectures have similar computational demands as their baselines. A simple procedure of replacing the 10-th and 16-th inverted bottleneck blocks with lightweight lambda blocks in the MobileNet-v2 architecture reduces parameters and flops by ∼10% while improving ImageNet accuracy by 0.6%. This suggest that lambda layers may be well suited for use in resource constrained scenarios such as embedded vision applications (Howard et al., 2017; Sandler et al., 2018; Howard et al., 2019).

| Architecture | Params (M) | FLOPS (M) | top-1 |
|---|---|---|---|
| MobileNet-v2 | 3.50 | 603 | 72.7 |
| MobileNet-v2 with 2 lightweight lambda blocks | **3.21** | **563** | **73.3** |

Table 17: **Lambda layers improve ImageNet accuracy in a resource-constrained scenario.** Replacing the 10-th and 16-th inverted bottleneck blocks with lightweight lambda blocks in the MobileNet-v2 architecture reduces parameters and flops by ∼10% while improving ImageNet accuracy by 0.6%.

# F    EXPERIMENTAL DETAILS

## F.1    DETAILED LAMBDARESNETS RESULTS

| Depth | Image size | Latency (s) | Supervised top-1 | Pseudo-labels top-1 |
|-------|-----------|-------------|------------------|---------------------|
| 50  | 128 | 0.058 | 77.4 | 82.1 |
| 50  | 160 | 0.089 | 79.2 | 83.4 |
| 101 | 160 | 0.14  | 80.8 | 84.7 |
| 101 | 192 | 0.20  | 81.9 | 85.4 |
| 152 | 192 | 0.28  | 82.5 | 86.1 |
| 152 | 224 | 0.38  | 83.2 | 86.5 |
| 152 | 256 | 0.49  | 83.8 | - |
| 152 | 288 | 0.63  | -    | 86.7 |
| 270 | 256 | 0.91  | 84.2 | - |
| 350 | 256 | 1.16  | 84.4 | - |
| 350 | 288 | 1.48  | 84.5 | - |
| 350 | 320 | 1.91  | 84.7 | - |
| 420 | 320 | 2.25  | 84.9 | - |

Table 18: **Detailed LambdaResNets results**. Latency refers to the time per training step for a batch size of 1024 on 8 TPU-v3 cores using `bfloat16` activations.

## F.2    ARCHITECTURAL DETAILS

**Lambda layer implementation details**    Unless specified otherwise, all lambda layers use query depth $|k|$=16, $|h|$=4 heads and intra-depth $|u|$=1. The *position* lambdas are generated with local contexts of size $|m|$=23x23 and the *content* lambdas with the global context using the einsum implementation as described in Figure 3. Local positional lambdas can be implemented interchangeably with the lambda convolution or by using the *global* einsum implementation and masking the position embeddings outside of the local contexts (Figure 5). The latter can be faster but has higher FLOPS and memory footprint due to the $\Theta(knm)$ term (see Table 2). In our experiments, we use the convolution implementation only for input length $|n| > 85^2$ or intra-depth $|u| > 1$. When the intra-depth is increased to $|u| > 1$, we switch to the convolution implementation and reduce the scope size to $|m|$=7x7 to reduce flops.

Positional embeddings are initialized at random using the unit normal distribution $\mathcal{N}(0,1)$. We use fan-in initialization for the linear projections in the lambda layer. The projections to compute $K$ and $V$ are initialized at random with the $\mathcal{N}(0, |d|^{-1/2})$ distribution. The projection to compute $Q$ is initialized at random with the $\mathcal{N}(0, |kd|^{-1/2})$ distribution (this is similar to the *scaled* dot-product attention mechanism, except that the scaling is absorbed in the projection). We apply batch normalization on $Q$ and $V$ and the keys $K$ are normalized via a softmax operation.

**ResNets.**    We use the ResNet-v1 implementation and initialize the $\gamma$ parameter in the last batch normalization (Ioffe & Szegedy, 2015) layer of the bottleneck blocks to 0. Squeeze-and-Excitation layers employ a squeeze ratio of 4. Similarly to ResNet-RS (Bello et al., 2021), we use the ResNet-D (He et al., 2018) and additionally replace the max pooling layer in the stem by a strided 3x3 convolution. Our block allocation and scaling strategy (i.e. selected resolution as a function of model depth) also follow closely the scaling recommendations from ResNet-RS (Bello et al., 2021).

**LambdaResNets.**    We construct our LambdaResNets by replacing the spatial 3x3 convolutions in the bottleneck blocks of the ResNet-RS architectures by our proposed lambda layer, with the exception of the stem which is left unchanged. We apply 3x3 average-pooling with stride 2 after the lambda layers to downsample in place of the strided convolution. Lambda layers are uniformly spaced in the `c4` stage and all bottlenecks in `c5` use lambda layers. Table 19 presents the exact block configuration and the location of the lambda layers for our hybrid LambdaResNets. We do not use squeeze-and-excitation in the bottleneck blocks that employ a lambda layer instead of the standard 3x3 convolution.

| Model | Block Configuration | Lambda layers in `c4` |
|---|---|---|
| LambdaResNet-50 | `[3-4-6-3]` | 3 |
| LambdaResNet-101 | `[3-4-23-3]` | 6, 12, 18 |
| LambdaResNet-152 | `[3-8-36-3]` | 5, 10, 15, 20, 25, 30 |
| LambdaResNet-200 | `[3-24-36-3]` | 5, 10, 15, 20, 25, 30 |
| LambdaResNet-270 | `[4-29-53-4]` | 8, 16, 24, 32, 40, 48 |
| LambdaResNet-350 | `[4-36-72-4]` | 10, 20, 30, 40, 50, 60 |
| LambdaResNet-420 | `[4-44-87-4]` | 10, 20, 30, 40, 50, 60, 70, 80 |

Table 19: **Block configurations and lambda layers placement of LambdaResNets in the Pareto curves**. LambdaResNets use the block allocations from He et al. (2016); Bello et al. (2021).

### F.3 TRAINING DETAILS

**ImageNet training setups.** We consider two training setups for the ImageNet classification task. The 90 epochs training setup trains models for 90 epochs using standard preprocessing and allows for fair comparisons with classic works. The 350 epochs training setup trains models for 350 epochs using improved data augmentation and regularization and is closer to training methodologies used in modern works with state-of-the-art accuracies.

**Supervised ImageNet 90 epochs training setup with vanilla ResNet.** In the 90 epoch setup, we use the *vanilla* ResNet for fair comparison with prior works. We used the default hyperparameters as found in official implementations without doing additional tuning. All networks are trained end-to-end for 90 epochs via backpropagation using SGD with momentum 0.9. The batch size $B$ is 4096 distributed across 32 TPUv3 cores (Jouppi et al., 2017) and the weight decay is set to 1e-4. The learning rate is scaled linearly from 0 to 0.1B/256 for 5 epochs and then decayed using the cosine schedule (Loshchilov & Hutter, 2017). We use batch normalization with decay 0.9999 and exponential moving average with weight 0.9999 over trainable parameters and a label smoothing of 0.1. The input image size is set to 224x224. We use standard training data augmentation (random crops and horizontal flip with 50% probability).

Most works compared against in Table 3 use a similar training setup and also replace the 3x3 spatial convolutions in the ResNet architecture by their proposed methods. We note that Ramachandran et al. (2019) train for longer (130 epochs instead of 90) but do not use label smoothing which could confound our comparisons.

**Supervised ImageNet 350 epochs training setup.** Higher accuracies on ImageNet are commonly obtained by training longer with increased augmentation and regularization (Lee et al., 2020; Tan & Le, 2019). Similarly to Bello et al. (2021), the weiht decay is reduced to 4e-5 and we employ RandAugment (Cubuk et al., 2019) with 2 layers, dropout (Srivastava et al., 2014) and stochastic depth (Huang et al., 2016). See Table 20 for exact hyperparameters. All architectures are trained for 350 epochs with a batch size B of 4096 or 2048 distributed across 32 or 64 TPUv3 cores, depending on memory constraints.

We tuned our models using a held-out validation set comprising ∼2% of the ImageNet training set (20 shards out of 1024). We perform early stopping on the held-out validation set for the largest models, starting with LambdaResNet-350 at resolution 288x288, and simply report the final accuracies for the smaller models.

**Semi-supervised learning with pseudo-labels.** Our training setup closely follows the experimental setup from Xie et al. (2020). We use the same dataset of 130M filtered and balanced JFT images with pseudo-labels generated by an EfficientNet-L2 model with 88.4% ImageNet accuracy. Hyperparameters are the same as for the supervised ImageNet 350 epochs experiments.

**Latency measurements.** Figure 4 reports training latencies (i.e. time per training step) to process a batch of 1024 images on 8 TPUv3 cores using mixed precision training (i.e `bfloat16` activations). Training latency is originally measured on 8 TPUv3 cores, starting with a total batch size of 1024 (i.e. 128 per core) and dividing the batch size by 2 until it fits in memory. We then report the *normalized* latencies in Figure 4. For example, if latency was measured with a batch size of 512

| Depth | Image Size | RandAugment magnitude | Dropout | Stochastic depth rate |
|-------|-----------|----------------------|---------|----------------------|
| 50  | 128 | 10 | 0.2 | 0   |
| 50  | 160 | 10 | 0.2 | 0   |
| 101 | 160 | 10 | 0.3 | 0   |
| 101 | 192 | 15 | 0.2 | 0   |
| 152 | 192 | 15 | 0.3 | 0   |
| 152 | 224 | 15 | 0.3 | 0.1 |
| 152 | 256 | 15 | 0.3 | 0.1 |
| 152 | 288 | 15 | 0.3 | 0.1 |
| 270 | 256 | 15 | 0.3 | 0.1 |
| 350 | 256 | 15 | 0.3 | 0.2 |
| 350 | 288 | 15 | 0.3 | 0.2 |
| 350 | 320 | 15 | 0.3 | 0.2 |
| 420 | 320 | 15 | 0.3 | 0.2 |

Table 20: **Hyperparameters used to train LambdaResNets**. We train for 350 epochs with RandAugment, dropout and stochastic depth.

(instead of 1024), we normalize the reported latency by multiplying the measured latency by 2. Table 4, Table 12 and Table 13 report *inference* throughput on 8 TPUv3 cores using full precision (i.e. `float32` activations). Latency for ViT (Dosovitskiy et al., 2020) was privately communicated by the authors.

**FLOPS count.** We do not count zeroed out flops when computing positional lambdas with the einsum implementation from Figure 3. Flops count is highly dependent on the scope size which is rather large by default ($|m|$=23x23). In Table 10, we show that it is possible to significantly reduce the scope size and therefore FLOPS at a minimal degradation in performance.

**COCO object detection.** We employ the architecture from the improved ImageNet training setup as the backbone in the Mask-RCNN architecture. All models are trained on 1024x1024 images from scratch for 130k steps with a batch size of 256 distributed across 128 TPUv3 cores with synchronized batch normalization. We apply multi-scale jitter of [0.1, 2.0] during training. The learning rate is warmed up for 1000 steps from 0 to 0.32 and divided by 10 at steps 90, 95 and 97.5% of training. The weight decay is set to 4e-5.

**Mobilenet training setup.** All mobilenet architectures are trained for 350 epochs on Imagenet with standard preprocessing at 224x224 resolution. We use the same hyperparameters as Howard et al. (2019). More specifically, we use RMSProp with 0.9 momentum and a batch size of 4096 split across 32 TPUv3 cores. The learning rate is warmed up linearly to 0.1 and then multiplied by 0.99 every 3 epochs. We use a weight decay 1e-5 and dropout with drop probability of 0.2

