# OpenReview forum: "LambdaNetworks: Modeling long-range Interactions without Attention"
_ICLR.cc/2021/Conference — ICLR 2021 Spotlight_

### Official Review · AnonReviewer1 · 2020-10-26
**The ideas and results are interesting, but it lacks some important discussion and results.**

**Rating:** 6
**Confidence:** 3

**Review:**

Summary
1. This paper present a new method, the lambda layer, for capturing long-term dependency.
2. lambda layer summarizes the context into the fixed vector to reduce the computation burden
3. This paper validates the performance on image classification and object detection tasks.

Strengths
1. The lambda layer is a simple and effective method.
2. The idea of context summarization is interesting.
3. Lambda layer shows the meaningful performance gain (including memory and time efficiency).

Weaknesses
1. This paper says that the lambda layer is a general framework, but the results only include vision tasks. Furthermore, there lacks discussion or results for the auto-regressive tasks.
2. The comparison between the lambda layer and linear attention is not well discussed.

Questions and Additional Feedback
1. Why Q=XW_{Q} is |m|\times|k|\times|u| sized tensor in table 1?
2. The idea of reducing the complexity in the lambda layer seems similar to the “Transformers are rnns:” papers. This paper says that “We argue that such approaches may be overly restrictive and unnecessarily complex in trying to closely approximate an attention similarity kernel”. It is hard to agree with the sentence. What is the reason behind the sentence? Are there any empirical or theoretical results?
3. This paper says that the lambda layer is “a general framework” and “versatile”. However, all of the experiments are related to the vision tasks. Are there results that related to natural language or time-series?
4. I wonder how lambda layers work for auto-regressive tasks. There is some description of auto-regressive training in the appendix, but it is not enough.
5. What is the advantage of lambda layers over attention? According to this paper, the only shortage of attention is an expensive computation. However, there are several linear time attention research. I suggest that this paper includes additional qualitative and quantitative analysis to compare the attention and linear time attention.

Typos
1. signifcantly => significantly
2. as the the queries => as the queries
3. formluations => formulations

Comments after the rebuttal
The author's response resolves my concern in part, and I will keep my positive score.

---

> ### Author Response · Authors · 2020-11-13
> **Initial reply to Reviewer 1**
>
> We thank the reviewer for a thoughtful review and constructive feedback. See our replies below.
>
> 1. _"Why are lambda layers presented as a versatile/general framework when the paper only includes vision experiments?"_
>     - "versatile" specifically refers to the ability to "model content and position-based interactions in global, local and masked contexts". The complete sentence is "Lambda layers are versatile and may be implemented to model content and position-based interactions in global, local or masked contexts".
>     - We believe that lambda layers will be applied other modalities as no assumption is specific to vision.
>     - _We will update the tone of the text to address the reviewer's concern._
>
> 2. _"Additional qualitative/quantitative comparison between lambda layers and linear attention is required [...] What is the advantage of lambda layers over attention, given that linear attention also has reduced complexity?"_
>     - __Linear attention methods do not model (translation equivariant) position-based interactions__. See Related Work: "Closest to our work are channel, spatial and linear attention mechanisms which can be cast as less flexible instances of _content-only_ lambda interactions. Lambda layers formalize and extend such approaches to __consider both content-based _and position-based interactions_, which enables their use as a stand-alone layer on highly structured inputs such as images__"
>     - __Table 3 readily includes comparisons against linear attention methods__ ("double attention" and "efficient attention" are linear attention methods). _Double attention and efficient attention do not model position-based interactions and therefore must be complemented with regular convolutions. The lambda layers outperform both linear attention formulations._
>     - Finally, we note that the lambda layer outperforms (local) self-attention when controlling for the scope size (e.g. 78.1% vs 77.4% for scope size $|m|$=7x7), demonstrating that _the benefits of lambda layers go beyond improved speed and scalability_ (see last sentence of "model ablations" in section 5).
>     - We will update the text to highlight these results.
>
> 3. "Further explanation is required to argue that factorizations of the attention kernel may be overly _restrictive_ and _unnecessarily complex_ in trying to closely approximate an attention similarity kernel”.
>     - _overly restrictive:_ The attention kernels in these works are approximated as $attn(q,k) \sim \phi(q)^T\phi(k)$ for some function $\phi$. $\phi$ is also sometimes constrained to be positive since the softmax attention kernel is positive (e.g. the Transformers are RNNs paper). The $\phi(q)^T\phi(k)$ form precludes the use of different normalization strategies on $q$ and $k$ which we find useful (Table 10 in Appendix E.3). Additionally, early experiments (not yet reported in the draft) showed that not applying a softmax to the queries is helpful. This means that (when $\phi$ is the softmax) $q^T\phi(k)$ (as done in lambda layers) outperforms the $\phi(q)^T\phi(k)$ attention approximation.  We refer to this in Related Work: "_Rather than attempting to closely approximate attention maps as is the case in linear attention formulations, the lambda abstraction shifts the focus to the design of efficient contextual lambda functions. This leads to [...] more flexible normalization schemes._"
>     - _unnecessarily complex:_ Another line of linear attention works consists in using random/orthogonal/Fourier features (e.g. [Random Feature Attention](https://openreview.net/forum?id=QtTKTdVrFBB) and [Rethinking Attention with Performers](https://openreview.net/forum?id=Ua6zuk0WRH)). While their results are promising, the theory behind these works is rather complex.
>     - We will update the draft to clarify this point.
>
> 4. Shape of $Q$?
>     - This is a mistake in the draft, we will correct in the next revision.
>
> 5. What about auto-regressive tasks?
>     - We include a discussion about auto-regressive training in the Appendix B and leave experiments for future work. We believe the current set of experiments (image classification across different model scales, object detection, instance segmentation, ablations) is already sufficient.

---

### Official Review · AnonReviewer4 · 2020-10-28
**A good module but incremental technical novelty compared with self-attention.**

**Rating:** 6
**Confidence:** 3

**Review:**

This work proposes a lambda layer to capture long-range context. To address the issue of heavy memory-cost in self-attention, the proposed lambda layers transform context aggregation in self-attention into individual linear functions. The system-level performance is good on classification, detection and instance segmentation.

Strengths:

->Less memory and parameters, lower complexity than self-attention.

->The performance is consistently good on classification, detection and instance segmentation.


Weaknesses:

->The main concern is technical novelty. Though interesting decomposition of query-context is introduced by linear functions, the overall design is still follow self-attention, the new part is incremental.

->Some notations are missed in Table 1. Better to involve all the used notations in the Table.

->The Lambda convolution performs local context, how about non-local context based on the proposed lambda function

->What is SE in Table 7? The full name should be used before using its acronym.

---

> ### Author Response · Authors · 2020-11-12
> **Initial reply to Reviewer 4**
>
> We thank the reviewer for their review and feedback. See our replies below.
>
> 1. "The main concern is lack of technical novelty"
>     - We respectfully but strongly disagree that the work is incremental. __We follow the terminology of attention to ease readability and highlight differences but the design doesn't follow self-attention__.
>     - In section 2, _we motivate lambda layers as an alternative and show that layers capturing interactions between inputs and their contexts can either contract the query depth first (attention) or the context length first (lambda layers)_.
>     - Additionally, we propose _multiquery lambdas to reduce complexity_ (in contrast, _multiquery /head attention increases representational power_) and introduce a _local implementation using convolution kernels_ (in contrast, _local self-attention materializes overlapping patches of query and memory blocks_)
>     - __We do not believe that a hypothetical lack of technical novelty should be an issue, especially in light of the strong empirical results__ (reduced memory requirements, ease of implementation, state-of-the-art speed-accuracy Pareto curve, improved params/flops efficiency compared to EfficientNets, improvements in detection/segmentation).
>
> 2. "Missed notations".
>     - We will update Table 1 to include all notations.
>
> 3. "What about non-local contexts based on the proposed lambda function?"
>     - In most experiments, we actually use the (global) einsum implementation from Equation 4. We study the importance of scope size in Table 9 (Appendix E.2). Since we find no benefits from using global contexts for position-based interactions, we mask interactions outside a scope of size |m|=23x23 (which is rather large). This is a choice, rather than a constraint of the method. This is currently explained in "Lambda layer implementation details" in Appendix D but we will update the main text to clarify this confusion.
>     - Finally, note that the content lambda uses global contexts.
>
> 4. "What is SE in Table 7?"
>     - SE refers to the popular squeeze-and-excitation operator (a.k.a channel attention). It also appears in Figure 2. We will update the draft to clarify this.

---

### Official Review · AnonReviewer3 · 2020-10-28
**Official Blind Review #3**

**Rating:** 6
**Confidence:** 4

**Review:**

This paper proposes a novel lambda layer to capture long-range interactions by transforming available contexts into linear functions, termed lambdas and applying these linear functions to each input separately. The proposed Lambda Network achieves good performances on ImageNet Classification, COCO object detection and instance segmentation tasks. The proposed lambda convolution is much more dense than the attention-based layer thus reducing parameters and complexity. However there are still several weaknesses in this paper. 1) Generalization of the proposed lambda convolution layer. For example, how about the performance of the lambda layer when combined with the lighter convolutional networks, e.g. mobilenet ? How about the performance when much deeper networks for the highest performance?  2)The source code is suggested to be released for more details. 3) Check the typos in the paper.

---

> ### Author Response · Authors · 2020-11-12
> **Initial reply to Reviewer 3**
>
> We thank the reviewer for a thoughtful review and constructive feedback. See our replies below.
>
> 1. "Generalization of the proposed lambda convolution layer (e.g. MobileNets and high-performing deep networks)".
>     - MobileNet: We will run experiments with lambda layers in a mobile-constrained setting.
>     - High-performing deep networks: _Figure 2, Table 5 and Table 6 already present results for  high-performing deep networks_. __Most notably, our deepest LambdaResNet reaches state-of-the-art accuracy of 84.8% and outperforms [EfficientNet-B7](https://github.com/tensorflow/tpu/tree/master/models/official/efficientnet) while being much faster (see Figure 2)__
> 2. "Source code is suggested to be released for more details"
>     - Training code and checkpoints will be open-sourced shortly as said in the draft (first paragraph of Experiments section).
> 3. "Check typos"
>     - Typos will be fixed in the revised version.
>
> __Does the reviewer have any other concerns that can be addressed besides mobilenet experiments?__

---

### Official Review · AnonReviewer2 · 2020-10-29
**Explanation can be further improved.**

**Rating:** 6
**Confidence:** 3

**Review:**

This paper presents an efficient method to model long-range interaction. The proposed lambda layer removes the nonlinearity of the original attention operation and makes the matrix multiplication independent of the context, hence skipping expensive computation and storage of large attention maps. Two kinds of lambda functions in lambda layer, i.e., content lambda and position lambda,  allows the model to capture both dense content and long-range interaction.  In addition, the lambda layer can be further extended to working with local context and to being more efficient by docomposing a query into multiple short ones. Its effectivess has been demonstrated on extensive experiments on different backbone network architectures and tasks. Its speed-accuracy tradeoff perform very favorably against SOTA methods.

However, there are still several issues to be addressed.
1. This paper is not easy to follow. There are too many symbols and several of them are not explained. Besides, the organization of the paper can also be further improved.

2. Some typos with the paper. E.g., In Table 1, tensor Q should have shape of nxkxu instead of mxkxu; lambda_n in eq. (2) should be transposed;

3. In section 3.2, it is not clear how symbol u comes and what it means.

4. Why are d and h removed from complexity analaysis in Table 4?

5. Although authors explain its value to long sequences and high-resolution images, there is no experiment on the corresponding tasks such as long sequence language translation or high-resolution pixel-level prediction tasks.

---

> ### Author Response · Authors · 2020-11-12
> **Initial reply to Reviewer 2**
>
> We thank the reviewer for a thoughtful review and constructive feedback. See our replies below.
>
> 1. & 2. _Typos and readability_:
>     - We will fix typos and further improve readability in the next revision.
> 3. _What is $u$ and where is it introduced?_:
>     - The intra-depth $u$ is a hyperparameter that controls the size of positional embeddings and keys/values _independently of the query depth_. It can increase representational power as seen in Table 3. It is introduced in Table 1 before section 3.2. We will update the main text to highlight this.
> 4.  _"Why are d and h removed from complexity analaysis in Table 4?"_:
>     - The exact spatial complexity for the lambda layer is O($knm + bnkd/h$) as shown in Table 2. The activations in the neural network are O($bnd$) which is similar to O($bnkd/h$) in practice. Therefore, only the $nm$ term may be problematic and we remove the $bnkd/h$ term for readability.
> 5.  _"No experiments on corresponding high-resolution (pixel-level) prediction tasks."_:
>     - Table 7 presents results on an __instance segmentation task (per-pixel prediction) on 1024x1024 inputs__.
>     - Additionally, Table 4 shows that __for the standard ImageNet setup (usually considered high-resolution compared to datasets like CIFAR for example), lambda layers are the only alternative that can model global content-based and position-based interactions.__

---

### Official Review · AnonReviewer5 · 2020-11-10
**A promising new attention-based layer studied on vision tasks**

**Rating:** 8
**Confidence:** 4

**Review:**

This paper presents a local attention + relative positional encoding type of network suited for image classification and detection tasks.

The first focus of the approach is to make attention scale to (2D) images. Vanilla (global) self-attention with an input of size $n$ (e.g. n=224x224 resized ImageNet) and a context size of $m$ (m=n if global) costs $nm$ memory. They decompose their approximation in two parts, the content attention and the positional embedding (which requires global attention). For the (dense) content part, in the same vein as multiple "linear attention" approximations (e.g. Linformer, Wang et al. 2020) they make this attention $nk$ with $k$ independent of $m$ and much smaller. For the (relative, this translation equivariant) positional embedding, the space cost is still $nm$, but doesn't depend on the image, so this factorization into content + position is beneficial for larger batch sizes.

Another contribution of this paper is to study the convolutional variant, so called "lambda convolutions" (strictly local relative position embedding) by setting the weights of the convolution dynamically based on the relative positional embeddings, and which can effectively reuse optimized [T|G]PU convolution kernels

They also break down the query in "multiquery lambdas" (followed by concatenation) to reduce the computational cost (as in grouped convolutions).
Finally, they construct LambdaResNets by hybridation with vanilla ResNets where they replace any (see Table 12 for full results) of the convolution layers by lambdas for a parameters/throughput/accuracy trade-off.

They perform experiments on ImageNet (classification) and COCO (detection, with Mask-RCNN), which show competitive results: beating ResNets and relevant variants on size and accuracy, but not speed. LambdaResNets also beat EfficientNets on speed-accuracy on ImageNet accross the board.


Some limitations of the paper and/or method include:
- (minor) The related work (which is only really included in Appendix) does not discuss Zhao et al. 2020 (which is in Table 3).
- A lot of the good/important content is in the Appendix (e.g. Appendix E.1 / Table 8 showing that **the positional embedding is absolutely necessary for good performance while the content part is quite optional**; or experiments on the scope size from E.2 / Table 9).
- Their model can be seen (and indeed that is how they propose to implement it) as a kind of ConvNet with dynamically computed filters. Still, there is hope to recover long(-er than context size) range attention with multiple layers (as in ConvNets) and this is not studied/discussed.
- (minor) Speed is problematic in the current implementation (see Table 12 and 13 in Appendix), how much of it is due to [optimized kernels for ResNets vs. einsum implementaion] vs. necessary computational cost? There is a bit of a FLOPS comparison in Tables 6, 8, 9 and accompanying text, but practice (e.g. Table 4, 12, 13 throughput) vs. theoretical complexity is not discussed.
- A small caveat is that the space cost vs. global (self-) attention is only really advantageous for large batch sizes.


Overall, this is a good, readable, well studied (if one considers the appendix) paper on a promising new hybrid conv/attention layer that yields small and accurate models for computer vision core tasks (classification and detection).


Typo:
"=This section" in 3.2

---

> ### Author Response · Authors · 2020-11-12
> **Initial reply to Reviewer 5**
>
> We thank the reviewer for a thoughtful review and constructive feedback. See our replies below.
>
> "Beating ResNets and relevant variants on size and accuracy, but not speed": __LambdaResNets actually also improve on the speed-accuracy trade-off of baseline ResNets and variants (see Figure 2).__
>
> 1) _"No discussion on Zhao et al, 2020":_
>     - We will add a discussion of this paper in the next revision.
> 2) _"A lot of the good/important content is in the Appendix":_
>     - We will move some of these results to the main text in the next revision if the page limit enables it. Note that the importance of position-based interactions and scope size are mentioned throughout the main text (e.g . "In contrast, the position lambda encodes how to transform the query content  based on positions (n, m), enabling modeling structured inputs such images.", "and position-based interactions, which enables their use as a stand-alone layer on highly structured inputs such as images" and "images are highly structured, making position-based interactions crucial").
> 3) _"Model is implemented as ConvNet with dynamically computed filters [...] but discussion of local vs global interactions is lacking":_
>     - __In most experiments, we actually use the (global) einsum implementation from Equation 4, which we found faster than the lambda convolution (see Table 4). We find no benefits from using global contexts for position-based interactions (see Table 9) so we mask interactions outside  a scope of size |m|=23x23 (which is rather large). This is a choice, rather than a constraint of the method.__ This is currently explained in "Lambda layer implementation details" in Appendix D but will update the main text to clarify this confusion.
>     - The lambda convolution is mostly relevant for very high-resolution images (where even the reduced O(kn^2) spatial complexity becomes problematic) and must be used with smaller kernel sizes to be reasonably fast. Additionally, note that content-based interactions are global in all our experiments.
> 4) _Practical speed vs theoretical complexity (FLOPS) requires further discussion:_
>     - A perhaps counter-intuitive observation is that the lambda convolution (which has lower FLOPs) is slower than the global einsum implementation (which has higher FLOPs). Increasing the scope size increases FLOPs (see Table 9) but not latency when using the global einsum implementation instead of the lambda convolution. We will update the text to highlight these results and include additional discussion on speed vs FLOPs.
> 5) _"Space cost of lambda layers is only really advantageous for large batch sizes:"_
>     - We show in Table 4 that positional embeddings can be shared across lambda layers to further reduce memory requirements. __This leads to a much more advantageous complexity of O($kn^2$) for lambda layers compared to O($blhn^2$) for attention, even when batch size = 1__ (b: batch, l: number of layers, k: query depth, h: number of heads, n: input length).
>     - Additionally, batch sizes are quite relevant in practice, especially for training. _Smaller memory requirements benefit even small batch sizes since they i) leave space to scale the model further (along depth for example) and ii) can make models faster (less memory access, improved fragmentation)._
>     - Finally, we note that the lambda layer outperforms (local) self-attention when controlling for the scope size (e.g. 78.1% vs 77.4% for $|m|=$7x7), demonstrating that the benefits of lambda layers go beyond improved speed and scalability (see last sentence of "model ablations" in section 5)

---

### Decision · Program_Chairs · 2021-01-07
**Final Decision**

**Decision:**

Accept (Spotlight)

**Comment:**

This paper introduces LambdaNetworks, a new method to capture long range interactions in data (such as global context in images). The method is novel and simple, and the experimental results are strong, especially in terms of speed/accuracy tradeoffs. The paper is well written and easy to follow. For these reasons, I recommend to accept the paper.